# Riemannian Residual Neural Networks

**Isay Katsman**[*]
Yale University
isay.katsman@yale.edu

**Eric M. Chen**[*]**, Sidhanth Holalkere**[*]
Cornell University
{emc348, sh844}@cornell.edu

**Anna Asch**
Cornell University
aca89@cornell.edu

**Aaron Lou**
Stanford University
aaronlou@stanford.edu

**Ser-Nam Lim**[†]
University of Central Florida
sernam@ucf.edu

**Christopher De Sa**
Cornell University
cdesa@cs.cornell.edu

## Abstract

Recent methods in geometric deep learning have introduced various neural networks to operate over data that lie on Riemannian manifolds. Such networks are often necessary to learn well over graphs with a hierarchical structure or to learn over manifold-valued data encountered in the natural sciences. These networks are often inspired by and directly generalize standard Euclidean neural networks. However, extending Euclidean networks is difficult and has only been done for a select few manifolds. In this work, we examine the residual neural network (ResNet) and show how to extend this construction to general Riemannian manifolds in a geometrically principled manner. Originally introduced to help solve the vanishing gradient problem, ResNets have become ubiquitous in machine learning due to their beneficial learning properties, excellent empirical results, and easy-to-incorporate nature when building varied neural networks. We find that our Riemannian ResNets mirror these desirable properties: when compared to existing manifold neural networks designed to learn over hyperbolic space and the manifold of symmetric positive definite matrices, we outperform both kinds of networks in terms of relevant testing metrics and training dynamics.

## 1 Introduction

In machine learning, it is common to represent data as vectors in Euclidean space (i.e. $\mathbb{R}^n$). The primary reason for such a choice is convenience, as this space has a classical vectorial structure, a closed-form distance formula, and a simple inner-product computation. Moreover, the myriad existing Euclidean neural network constructions enable performant learning.

Despite the ubiquity and success of Euclidean embeddings, recent research [41] has brought attention to the fact that several kinds of complex data require manifold considerations. Such data are various and range from covariance matrices, represented as points on the manifold of symmetric positive definite (SPD) matrices [26], to angular orientations, represented as points on tori, found in the context of robotics [43]. However, generalizing Euclidean neural network tools to manifold structures such as these can be quite difficult in practice. Most prior works design network architectures for a specific manifold [11, 17], thereby inefficiently necessitating a specific design for each new manifold.

We address this issue by extending Residual Neural Networks [23] to Riemannian manifolds in a way that naturally captures the underlying geometry. We construct our network by parameterizing vector fields and leveraging geodesic structure (provided by the Riemannian $\exp$ map) to "add" the learned

---

[*] indicates equal contribution.
[†] Work done while at Meta AI.

vectors to the input points, thereby naturally generalizing a typical Euclidean residual addition. This process is illustrated in Figure 1. Note that this strategy is exceptionally natural, only making use of inherent geodesic geometry, and works generally for all smooth manifolds. We refer to such networks as Riemannian residual neural networks.

Though the above approach is principled, it is under-specified, as constructing an efficient learnable vector field for a given manifold is often nontrivial. To resolve this issue, we present a general way to induce a learnable vector field for a manifold $\mathcal{M}$ given only a map $f : \mathcal{M} \to \mathbb{R}^k$. Ideally, this map should capture intrinsic manifold geometry. For example, in the context of Euclidean space, this map could consist of a series of $k$ projections onto hyperplanes. There is a natural equivalent of this in hyperbolic space that instead projects to horospheres (horospheres correspond to hyperplanes in Euclidean space). More generally, we propose a feature map that once more relies only on geodesic information, consisting of projection to random (or learned) geodesic balls. This final approach provides a fully geometric way to construct vector fields, and therefore natural residual networks, for any Riemannian manifold.

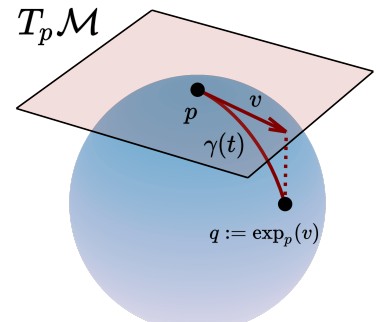

Figure 1: An illustration of a manifold-generalized residual addition. The traditional Euclidean formula $p \leftarrow p + v$ is generalized to $p \leftarrow \exp_p(v)$, where $\exp$ is the Riemannian exponential map. $\mathcal{M}$ is the manifold and $T_p\mathcal{M}$ is the tangent space at $p$.

After introducing our general theory, we give concrete manifestations of vector fields, and therefore residual neural networks, for hyperbolic space and the manifold of SPD matrices. We compare the performance of our Riemannian residual neural networks to that of existing manifold-specific networks on hyperbolic space and on the manifold of SPD matrices, showing that our networks perform much better in terms of relevant metrics due to their improved adherence to manifold geometry.

Our contributions are as follows:

1. We introduce a novel and principled generalization of residual neural networks to general Riemannian manifolds. Our construction relies only on knowledge of geodesics, which capture manifold geometry.

2. Theoretically, we show that our methodology better captures manifold geometry than pre-existing manifold-specific neural network constructions. Empirically, we apply our general construction to hyperbolic space and to the manifold of SPD matrices. On various hyperbolic graph datasets (where hyperbolicity is measured by Gromov $\delta$-hyperbolicity) our method considerably outperforms existing work on both link prediction and node classification tasks. On various SPD covariance matrix classification datasets, a similar conclusion holds.

3. Our method provides a way to directly vary the geometry of a given neural network without having to construct particular operations on a per-manifold basis. This provides the novel capability to directly compare the effect of geometric representation (in particular, evaluating the difference between a given Riemannian manifold $(\mathcal{M}, g)$ and Euclidean space $(\mathbb{R}^n, || \cdot ||_2)$) while fixing the network architecture.

## 2   Related Work

Our work is related to but distinctly different from existing neural ordinary differential equation (ODE) [9] literature as well a series of papers that have attempted generalizations of neural networks to specific manifolds such as hyperbolic space [17] and the manifold of SPD matrices [26].

### 2.1   Residual Networks and Neural ODEs

Residual networks (ResNets) were originally developed to enable training of larger networks, previously prone to vanishing and exploding gradients [23]. Later on, many discovered that by adding a learned residual, ResNets are similar to Euler's method [9, 21, 37, 45, 53]. More specifically, the ResNet represented by $\mathbf{h}_{t+1} = \mathbf{h}_t + f(\mathbf{h}, \theta_t)$ for $\mathbf{h}_t \in \mathbb{R}^D$ mimics the dynamics of the ODE defined by $\frac{d\mathbf{h}(t)}{dt} = f(\mathbf{h}(t), t, \theta)$. Neural ODEs are defined precisely as ODEs of this form, where

the local dynamics are given by a parameterized neural network. Similar to our work, Falorsi and Forré [15], Katsman et al. [29], Lou et al. [36], Mathieu and Nickel [38] generalize neural ODEs to Riemannian manifolds (further generalizing manifold-specific work such as Bose et al. [3], that does this for hyperbolic space). However, instead of using a manifold's vector fields to solve a neural ODE, we learn an objective by parameterizing the vector fields directly (Figure 2). Neural ODEs and their generalizations to manifolds parameterize a continuous collection of vector fields over time for a single manifold in a dynamic flow-like construction. Our method instead parameterizes a discrete collection of vector fields, entirely untethered from any notion of solving an ODE. This makes our construction a strict generalization of both neural ODEs and their manifold equivalents [15, 29, 36, 38].

## 2.2 Riemannian Neural Networks

Past literature has attempted generalizations of Euclidean neural networks to a number of manifolds.

**Hyperbolic Space** Ganea et al. [17] extended basic neural network operations (e.g. activation function, linear layer, recurrent architectures) to conform with the geometry of hyperbolic space through gyrovector constructions [51]. In particular, they use gyrovector constructions [51] to build analogues of activation functions, linear layers, and recurrent architectures. Building on this approach, Chami et al. [8] adapt these constructions to hyperbolic versions of the feature transformation and neighborhood aggregation steps found in message passing neural networks. Additionally, batch normalization for hyperbolic space was introduced in Lou et al. [35]; hyperbolic attention network equivalents were introduced in Gülçehre et al. [20]. Although gyrovector constructions are algebraic and allow for generalization of neural network operations to hyperbolic space and beyond, we note that they do not capture intrinsic geodesic geometry. In particular, we note that the gyrovector-based hyperbolic linear layer introduced in Ganea et al. [17] reduces to a Euclidean matrix multiplication followed by a learned hyperbolic bias addition (see Appendix D.2). Hence all non-Euclidean learning for this case happens through the bias term. In an attempt to resolve this, further work has focused on imbuing these neural networks with more hyperbolic functions [10, 49]. Chen et al. [10] notably constructs a hyperbolic residual layer by projecting an output onto the Lorentzian manifold. However, we emphasize that our construction is more general while being more geometrically principled as we work with fundamental manifold operations like the exponential map rather than relying on the niceties of Lorentz space.

Yu and De Sa [55] make use of randomized hyperbolic Laplacian features to learn in hyperbolic space. We note that the features learned are shallow and are constructed from a specific manifestation of the Laplace-Beltrami operator for hyperbolic space. In contrast, our method is general and enables non-shallow (i.e., multi-layer) feature learning.

**SPD Manifold** Neural network constructs have been extended to the manifold of symmetric positive definite (SPD) matrices as well. In particular, SPDNet [26] is an example of a widely adopted SPD manifold neural network which introduced SPD-specific layers analogous to Euclidean linear and ReLU layers. Building upon SPDNet, Brooks et al. [5] developed a batch normalization method to be used with SPD data. Additionally, López et al. [34] adapted gyrocalculus constructions used in hyperbolic space to the SPD manifold.

**Symmetric Spaces** Further work attempts generalization to symmetric spaces. Sonoda et al. [50] design fully-connected networks over noncompact symmetric spaces using particular theory from Helgason-Fourier analysis [25], and Chakraborty et al. [7] attempt to generalize several operations such as convolution to such spaces by adapting and developing a weighted Fréchet mean construction. We note that the Helgason-Fourier construction in Sonoda et al. [50] exploits a fairly particular structure, while the weighted Fréchet mean construction in Chakraborty et al. [7] is specifically introduced for convolution, which is not the focus of our work (we focus on residual connections).

Unlike any of the manifold-specific work described above, our residual network construction can be applied generally to any smooth manifold and is constructed solely from geodesic information.

## 3 Background

In this section, we cover the necessary background for our paper; in particular, we introduce the reader to the necessary constructs from Riemannian geometry. For a detailed introduction to Riemannian geometry, we refer the interested reader to textbooks such as Lee [32].

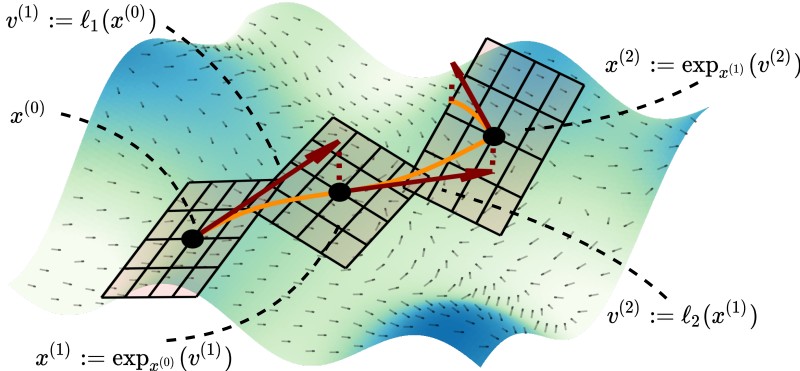

$v^{(1)} := \ell_1(x^{(0)})$

$x^{(0)}$

$x^{(2)} := \exp_{x^{(1)}}(v^{(2)})$

$v^{(2)} := \ell_2(x^{(1)})$

$x^{(1)} := \exp_{x^{(0)}}(v^{(1)})$

Figure 2: A visualization of a Riemannian residual neural network on a manifold $\mathcal{M}$. Our model parameterizes vector fields on a manifold. At each layer in our network, we take a step from a point in the direction of that vector field (brown), which is analogous to the residual step in a ResNet.

## 3.1 Riemannian Geometry

A topological manifold $(\mathcal{M}, g)$ of dimension $n$ is a locally Euclidean space, meaning there exist homeomorphic[1] functions (called "charts") whose domains both cover the manifold and map from the manifold into $\mathbb{R}^n$ (i.e. the manifold "looks like" $\mathbb{R}^n$ locally). A smooth manifold is a topological manifold for which the charts are not simply homeomorphic, but diffeomorphic, meaning they are smooth bijections mapping into $\mathbb{R}^n$ and have smooth inverses. We denote $T_p\mathcal{M}$ as the tangent space at a point $p$ of the manifold $\mathcal{M}$. Further still, a Riemannian manifold[2] $(\mathcal{M}, g)$ is an $n$-dimensional smooth manifold with a smooth collection of inner products $(g_p)_{p \in \mathcal{M}}$ for every tangent space $T_p\mathcal{M}$. The Riemannian metric $g$ induces a distance $d_g : \mathcal{M} \times \mathcal{M} \to \mathbb{R}$ on the manifold.

## 3.2 Geodesics and the Riemannian Exponential Map

**Geodesics** A geodesic is a curve of minimal length between two points $p, q \in \mathcal{M}$, and can be seen as the generalization of a straight line in Euclidean space. Although a choice of Riemannian metric $g$ on $\mathcal{M}$ appears to only define geometry locally on $\mathcal{M}$, it induces global distances by integrating the length (of the "speed" vector in the tangent space) of a shortest path between two points:

$$d(p, q) = \inf_{\gamma} \int_0^1 \sqrt{g_{\gamma(t)}(\gamma'(t), \gamma'(t))} \, dt \tag{1}$$

where $\gamma \in C^\infty([0, 1], \mathcal{M})$ is such that $\gamma(0) = p$ and $\gamma(1) = q$.

For $p \in \mathcal{M}$ and $v \in T_p\mathcal{M}$, there exists a unique geodesic $\gamma_v$ where $\gamma(0) = p$, $\gamma'(0) = v$ and the domain of $\gamma$ is as large as possible. We call $\gamma_v$ the maximal geodesic [32].

**Exponential Map** The Riemannian exponential map is a way to map $T_p\mathcal{M}$ to a neighborhood around $p$ using geodesics. The relationship between the tangent space and the exponential map output can be thought of as a local linearization, meaning that we can perform typical Euclidean operations in the tangent space before projecting to the manifold via the exponential map to capture the local on-manifold behavior corresponding to the tangent space operations. For $p \in \mathcal{M}$ and $v \in T_p\mathcal{M}$, the exponential map at $p$ is defined as $\exp_p(v) = \gamma_v(1)$.

One can think of $\exp$ as a manifold generalization of Euclidean addition, since in the Euclidean case we have $\exp_p(v) = p + v$.

---

[1]A homeomorphism is a continuous bijection with continuous inverse.
[2]Note that imposing Riemannian structure does not considerably limit the generality of our method, as any smooth manifold that is Hausdorff and second countable has a Riemannian metric [32].

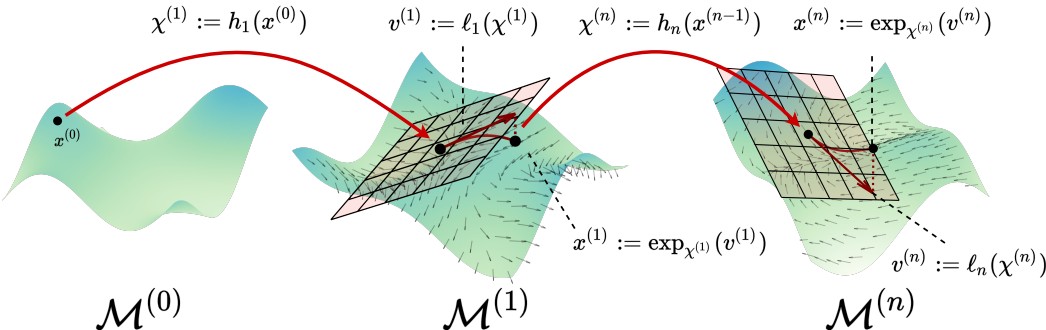

$$\chi^{(1)} := h_1(x^{(0)}) \qquad v^{(1)} := \ell_1(\chi^{(1)}) \qquad \chi^{(n)} := h_n(x^{(n-1)}) \qquad x^{(n)} := \exp_{\chi^{(n)}}(v^{(n)})$$

$$x^{(1)} := \exp_{\chi^{(1)}}(v^{(1)})$$

$$v^{(n)} := \ell_n(\chi^{(n)})$$

$$\mathcal{M}^{(0)} \qquad \mathcal{M}^{(1)} \qquad \mathcal{M}^{(n)}$$

Figure 3: An overview of our generalized Riemannian Residual Neural Network (RResNet) methodology. We start by mapping $x^{(0)} \in \mathcal{M}^{(0)}$ to $\chi^{(1)} \in \mathcal{M}^{(1)}$ using a base point mapping $h_1$. Then, using our paramterized vector field $\ell_i$, we compute a residual $v^{(1)} := \ell_1(\chi^{(1)})$. Finally, we project $v^{(1)}$ back onto the manifold using the Riemannian $\exp$ map, leaving us with $x^{(1)}$. This procedure can be iterated to produce a multi-layer Riemannian residual neural network that is capable of changing manifold representation on a per layer basis.

### 3.3 Vector Fields

Let $T_p\mathcal{M}$ be the tangent space to a manifold $\mathcal{M}$ at a point $p$. Like in Euclidean space, a vector field assigns to each point $p \in \mathcal{M}$ a tangent vector $X_p \in T_p\mathcal{M}$. A smooth vector field assigns a tangent vector $X_p \in T_p\mathcal{M}$ to each point $p \in \mathcal{M}$ such that $X_p$ varies smoothly in $p$.

**Tangent Bundle** The tangent bundle of a smooth manifold $\mathcal{M}$ is the disjoint union of the tangent spaces $T_p\mathcal{M}$, for all $p \in \mathcal{M}$, denoted by $T\mathcal{M} := \bigsqcup_{p \in \mathcal{M}} T_p\mathcal{M} = \bigsqcup_{p \in \mathcal{M}} \{(p, v) \mid v \in T_p\mathcal{M}\}$.

**Pushforward** A derivative (also called a *pushforward*) of a map $f : \mathcal{M} \to \mathcal{N}$ between two manifolds is denoted by $D_p f : T_p\mathcal{M} \to T_{f(p)}\mathcal{N}$. This is a generalization of the classical Euclidean Jacobian (since $\mathbb{R}^n$ is a manifold), and provides a way to relate tangent spaces at different points on different manifolds.

**Pullback** Given $\phi : \mathcal{M} \to \mathcal{N}$ a smooth map between manifolds and $f : \mathcal{N} \to \mathbb{R}$ a smooth function, the pullback of $f$ by $\phi$ is the smooth function $\phi^* f$ on $\mathcal{M}$ defined by $(\phi^* f)(x) = f(\phi(x))$. When the map $\phi$ is implicit, we simply write $f^*$ to mean the pullback of $f$ by $\phi$.

### 3.4 Model Spaces in Riemannian Geometry

The three Riemannian model spaces are Euclidean space $\mathbb{R}^n$, hyperbolic space $\mathbb{H}^n$, and spherical space $\mathbb{S}^n$, that encompass all manifolds with constant sectional curvature. Hyperbolic space manifests in several representations like the Poincaré ball, Lorentz space, and the Klein model. We use the Poincaré ball model for our Riemannian ResNet design (see Appendix A for more details on the Poincaré ball model).

### 3.5 SPD Manifold

Let $SPD(n)$ be the manifold of $n \times n$ symmetric positive definite (SPD) matrices. We recall from Gallier and Quaintance [16] that $SPD(n)$ has a Riemannian exponential map (at the identity) equivalent to the matrix exponential. Two common metrics used for $SPD(n)$ are the log-Euclidean metric [16], which induces a flat structure on the matrices, and the canonical affine-invariant metric [12, 42], which induces non-constant negative sectional curvature. The latter gives $SPD(n)$ a considerably less trivial geometry than that exhibited by the Riemannian model spaces [2] (see Appendix A for more details on $SPD(n)$).

## 4 Methodology

In this section, we provide the technical details behind Riemannian residual neural networks.

### 4.1 General Construction

We define a **Riemannian Residual Neural Network** (RResNet) on a manifold $\mathcal{M}$ to be a function $f : \mathcal{M} \to \mathcal{M}$ defined by

$$f(x) := x^{(m)} \tag{2}$$

$$x^{(0)} := x \tag{3}$$

$$x^{(i)} := \exp_{x^{(i-1)}}(\ell_i(x^{(i-1)})) \tag{4}$$

for $x \in \mathcal{M}$, where $m$ is the number of layers and $\ell_i : \mathcal{M} \to T\mathcal{M}$ is a neural network-parameterized vector field over $\mathcal{M}$. This residual network construction is visualized for the purpose of intuition in Figure 2. In practice, parameterizing a function from an abstract manifold $\mathcal{M}$ to its tangent bundle is difficult. However, by the Whitney embedding theorem [33], we can embed $\mathcal{M} \hookrightarrow \mathbb{R}^D$ smoothly for some dimension $D \geq \dim \mathcal{M}$. As such, for a standard neural network $n_i : \mathbb{R}^D \to \mathbb{R}^D$ we can construct $\ell_i$ by

$$\ell_i(x) := \mathrm{proj}_{T_x\mathcal{M}}(n_i(x)) \tag{5}$$

where we note that $T_x\mathcal{M} \subset \mathbb{R}^D$ is a linear subspace (making the projection operator well defined). Throughout the paper we call this the embedded vector field design[3]. We note that this is the same construction used for defining the vector field flow in Lou et al. [36], Mathieu and Nickel [38], Rozen et al. [44].

We also extend our construction to work in settings where the underlying manifold changes from layer to layer. In particular, for a sequence of manifolds $\mathcal{M}^{(0)}, \mathcal{M}^{(1)}, \ldots, \mathcal{M}^{(m)}$ with (possibly learned) maps $h_i : \mathcal{M}^{(i-1)} \to \mathcal{M}^{(i)}$, our Riemannian ResNet $f : \mathcal{M}^{(0)} \to \mathcal{M}^{(m)}$ is given by

$$f(x) := x^{(m)} \tag{6}$$

$$x^{(0)} := x \tag{7}$$

$$x^{(i)} := \exp_{h_i(x^{(i-1)})}(\ell_i(h_i(x^{(i-1)}))) \forall i \in [m] \tag{8}$$

with functions $\ell_i : \mathcal{M}^{(i)} \to T\mathcal{M}^{(i)}$ given as above. This generalization is visualized in Figure 3. In practice, our $\mathcal{M}^{(i)}$ will be different dimensional versions of the same geometric space (e.g. $\mathbb{H}^n$ or $\mathbb{R}^n$ for varying $n$). If the starting and ending manifolds are the same, the maps $h_i$ will simply be standard inclusions. When the starting and ending manifolds are different, the $h_i$ may be standard neural networks for which we project the output, or the $h_i$ may be specially design learnable maps that respect manifold geometry. As a concrete example, our $h_i$ for the SPD case map from an SPD matrix of one dimension to another by conjugating with a Stiefel matrix [26]. Furthermore, as shown in Appendix D, our model is equivalent to the standard ResNet when the underlying manifold is $\mathbb{R}^n$.

**Comparison with Other Constructions** We discuss how our construction compares with other methods in Appendix E, but here we briefly note that unlike other methods, our presented approach is fully general and better conforms with manifold geometry.

### 4.2 Feature Map-Induced Vector Field Design

Most of the difficulty in application of our general vector field construction comes from the design of the learnable vector fields $\ell_i : \mathcal{M}^{(i)} \to T\mathcal{M}^{(i)}$. Although we give an embedded vector field design above, it is not very principled geometrically. We would like to considerably restrict these vector fields so that their range is informed by the underlying geometry of $\mathcal{M}$. For this, we note that it is possible to induce a vector field $\xi : \mathcal{M} \to T\mathcal{M}$ for a manifold $\mathcal{M}$ with any smooth map $f : \mathcal{M} \to \mathbb{R}^k$. In practice, this map should capture intrinsic geometric properties of $\mathcal{M}$ and can be viewed as a feature map, or de facto linearization of $\mathcal{M}$. Given an $x \in \mathcal{M}$, we need only pass $x$ through $f$ to get its feature representation in $\mathbb{R}^k$, then note that since:

$$D_p f : T_p\mathcal{M} \to T_{f(p)}\mathbb{R}^k,$$

we have an induced map:

$$(D_p f)^* : (T_{f(p)}\mathbb{R}^k)^* \to (T_p\mathcal{M})^*,$$

where $(D_p f)^*$ is the pullback of $D_p f$. Note that $T_p\mathbb{R}^k \cong \mathbb{R}^k$ and $(\mathbb{R}^k)^* \cong \mathbb{R}^k$ by the dual space isomorphism. Moreover $(T_p\mathcal{M})^* \cong T_p\mathcal{M}$ by the tangent-cotangent space isomorphism [33]. Hence, we have the induced map:

$$(D_p f)_r^* : \mathbb{R}^k \to T_p\mathcal{M},$$

---

[3]Ideal vector field design is in general nontrivial and the embedded vector field is not a good choice for all manifolds (see Appendix B).

obtained from $(D_p f)^*$, simply by both precomposing and postcomposing the aforementioned iso-morphisms, where relevant. $(D_p f)^*_r$ provides a natural way to map from the feature representation to the tangent bundle. Thus, we may view the map $\ell_f : \mathcal{M} \to T\mathcal{M}$ given by:

$$\ell_f(x) = (D_x f)^*_r(f(x))$$

as a deterministic vector field induced entirely by $f$.

**Learnable Feature Map-Induced Vector Fields** We can easily make the above vector field construction learnable by introducing a Euclidean neural network $n_\theta : \mathbb{R}^k \to \mathbb{R}^k$ after $f$ to obtain $\ell_{f,\theta}(x) = (D_x f)^*(n_\theta(f(x)))$.

**Feature Map Design** One possible way to simplify the design of the above vector field is to further break down the map $f : \mathcal{M} \to \mathbb{R}^k$ into $k$ maps $f_1, \ldots, f_k : \mathcal{M} \to \mathbb{R}$, where ideally, each map $f_i$ is constructed in a similar way (e.g. performing some kind of geometric projection, where the $f_i$ vary only in terms of the specifying parameters). As we shall see in the following subsection, this ends up being a very natural design decision.

In what follows, we shall consider only smooth feature maps $f : \mathcal{M} \to \mathbb{R}^k$ induced by a single parametric construction $g_\theta : \mathcal{M} \to \mathbb{R}$, i.e. the $k$ dimensions of the output of $f$ are given by different choices of $\theta$ for the same underlying feature map[4]. This approach also has the benefit of a very simple interpretation of the induced vector field. Given feature maps $g_{\theta_1}, \ldots, g_{\theta_k} : \mathcal{M} \to \mathbb{R}$ that comprise our overall feature map $f : \mathcal{M} \to \mathbb{R}^k$, our vector field is simply a linear combination of the maps $\nabla g_{\theta_i} : \mathcal{M} \to T\mathcal{M}$. If the $g_{\theta_i}$ are differentiable with respect to $\theta_i$, we can even learn the $\theta_i$ themselves.

### 4.2.1 Manifold Manifestations

In this section, in an effort to showcase how simple it is to apply our above theory to come up with natural vector field designs, we present several constructions of manifold feature maps $g_\theta : \mathcal{M} \to \mathbb{R}$ that capture the underlying geometry of $\mathcal{M}$ for various choices of $\mathcal{M}$. Namely, in this section we provide several examples of $f : \mathcal{M} \to \mathbb{R}$ that induce $\ell_f : \mathcal{M} \to T\mathcal{M}$, thereby giving rise to a Riemannian neural network by Section 4.1.

**Euclidean Space** To build intuition, we begin with an instructive case. We consider designing a feature map for the Euclidean space $\mathbb{R}^n$. A natural design would follow simply by considering hyperplane projection. Let a hyperplane $w^T x + b = 0$ be specified by $w \in \mathbb{R}^n, b \in \mathbb{R}$. Then a natural feature map $g_{w,b} : \mathbb{R}^n \to \mathbb{R}$ parameterized by the hyperplane parameters is given by hyperplane projection [14]: $g_{w,b}(x) = \frac{|w^T x + b|}{||w||_2}$.

**Hyperbolic Space** We wish to construct a natural feature map for hyperbolic space. Seeking to follow the construction given in the Euclidean context, we wish to find a hyperbolic analog of hyperplanes. This is provided to us via the notion of horospheres [24]. Illustrated in Figure 4, horospheres naturally generalize hyperplanes to hyperbolic space. We specify a horosphere in the Poincaré ball model of hyperbolic space $\mathbb{H}^n$ by a point of tangency $\omega \in \mathbb{S}^{n-1}$ and a real value $b \in \mathbb{R}$. Then a natural feature map $g_{\omega,b} : \mathbb{H}^n \to \mathbb{R}$ parameterized by the horosphere parameters would be given by horosphere projection [4]:

$g_{\omega,b}(x) = -\log\left(\frac{1-||x||_2^2}{||x-\omega||_2^2}\right) + b.$

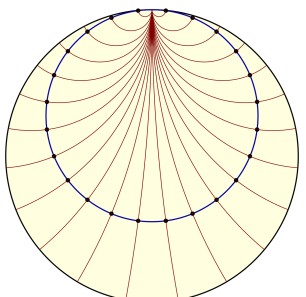

Figure 4: Example of a horosphere in the Poincaré ball representation of hyperbolic space. In this particular two-dimensional case, the hyperbolic space $\mathbb{H}_2$ is visualized via the Poincaré disk model, and the horosphere, shown in blue, is called a horocycle.

**Symmetric Positive Definite Matrices** The manifold of SPD matrices is an example of a manifold where there is no innate representation of a hyperplane. Instead, given $X \in SPD(n)$, a reasonable feature map $g_k : SPD(n) \to \mathbb{R}$, parameterized by $k$, is to map $X$ to its $k$th largest eigenvalue: $g_k(X) = \lambda_k$.

---

[4]We use the term "feature map" for both the overall feature map $f : \mathcal{M} \to \mathbb{R}^k$ and for the inducing construction $g_\theta : \mathcal{M} \to \mathbb{R}$. This is well-defined since in our work we consider only feature maps $f : \mathcal{M} \to \mathbb{R}^k$ that are induced by some $g_\theta : \mathcal{M} \to \mathbb{R}$.

**General Manifolds** For general manifolds there is no perfect analog of a hyperplane, and hence there is no immediately natural feature map. Although this is the case, it is possible to come up with a reasonable alternative. We present such an alternative in Appendix B.4 together with pertinent experiments.

***Example: Euclidean Space*** One motivation for the vector field construction $\ell_f(x) = (D_x f)_r^*(f(x))$ is that in the Euclidean case, $\ell_f$ will reduce to a standard linear layer (because the maps $f$ and $(D_x f)^*$ are linear), which, in combination with the Euclidean $\exp$ map, will produce a standard Euclidean residual neural network.

Explicitly, for the Euclidean case, note that our feature map $f : \mathbb{R}^n \to \mathbb{R}^k$ will, for example, take the form $f(x) = Wx, W \in \mathbb{R}^{k \times n}$ (here we have $b = 0$ and $W$ has normalized row vectors). Then note that we have $Df = W$ and $(Df)^* = W^T$. We see for the standard feature map-based construction, our vector field $\ell_f(x) = (D_x f)^*(f(x))$ takes the form $\ell_f(x) = W^T W x$.

For the learnable case (which is standard for us, given that we learn Riemannian residual neural networks), when the manifold is Euclidean space, the general expression $\ell_{f,\theta}(x) = (D_x f)^*(n_\theta(f(x)))$ becomes $\ell_{f,\theta}(x) = W^T n_\theta(Wx)$. When the feature maps are trivial projections (onto axis-aligned hyperplanes), we have $W = I$ and $\ell_{f,\theta}(x) = n_\theta(x)$. Thus our construction can be viewed as a generalization of a standard neural network.

| | Dataset | Disease | | Airport | | PubMed | | CoRA | |
| | Hyperbolicity | $\delta = 0$ | | $\delta = 1$ | | $\delta = 3.5$ | | $\delta = 11$ | |
| | Task | LP | NC | LP | NC | LP | NC | LP | NC |
|---|---|---|---|---|---|---|---|---|---|
| Shallow | Euc | $59.8_{\pm2.0}$ | $32.5_{\pm1.1}$ | $92.0_{\pm0.0}$ | $60.9_{\pm3.4}$ | $83.3_{\pm0.1}$ | $48.2_{\pm0.7}$ | $82.5_{\pm0.3}$ | $23.8_{\pm0.7}$ |
| | Hyp [41] | $63.5_{\pm0.6}$ | $45.5_{\pm3.3}$ | $94.5_{\pm0.0}$ | $70.2_{\pm0.1}$ | $87.5_{\pm0.1}$ | $68.5_{\pm0.3}$ | $87.6_{\pm0.2}$ | $22.0_{\pm1.5}$ |
| | Euc-Mixed | $49.6_{\pm1.1}$ | $35.2_{\pm3.4}$ | $91.5_{\pm0.1}$ | $68.3_{\pm2.3}$ | $86.0_{\pm1.3}$ | $63.0_{\pm0.3}$ | $84.4_{\pm0.2}$ | $46.1_{\pm0.4}$ |
| | Hyp-Mixed | $55.1_{\pm1.3}$ | $56.9_{\pm1.5}$ | $93.3_{\pm0.0}$ | $69.6_{\pm0.1}$ | $83.8_{\pm0.3}$ | $\mathbf{73.9}_{\pm0.2}$ | $85.6_{\pm0.5}$ | $45.9_{\pm0.3}$ |
| NN | MLP | $72.6_{\pm0.6}$ | $28.8_{\pm2.5}$ | $89.8_{\pm0.5}$ | $68.6_{\pm0.6}$ | $84.1_{\pm0.9}$ | $72.4_{\pm0.2}$ | $83.1_{\pm0.5}$ | $51.5_{\pm1.0}$ |
| | HNN [17] | $75.1_{\pm0.3}$ | $41.0_{\pm1.8}$ | $90.8_{\pm0.2}$ | $80.5_{\pm0.5}$ | $\mathbf{94.9}_{\pm0.1}$ | $69.8_{\pm0.4}$ | $\mathbf{89.0}_{\pm0.1}$ | $\mathbf{54.6}_{\pm0.4}$ |
| | **RResNet Horo** | $\mathbf{98.4}_{\pm0.3}$ | $\mathbf{76.8}_{\pm2.0}$ | $\mathbf{95.2}_{\pm0.1}$ | $\mathbf{96.9}_{\pm0.3}$ | $95.0_{\pm0.3}$ | $72.3_{\pm1.7}$ | $86.7_{\pm6.3}$ | $52.4_{\pm5.5}$ |

Table 1: Above we give graph task results for RResNet Horo compared with several non-graph-based neural network baselines (baseline methods and metrics are from Chami et al. [8]). Test ROC AUC is the metric reported for link prediction (LP) and test F1 score is the metric reported for node classification (NC). Mean and standard deviation are given over five trials. Note that RResNet Horo considerably outperforms HNN on the most hyperbolic datasets, performing worse and worse as hyperbolicity increases, to a more extreme extent than previous methods that do not adhere to geometry as closely (this is expected).

# 5 Experiments

In this section, we perform a series of experiments to evaluate the effectiveness of RResNets on tasks arising on different manifolds. In particular, we explore hyperbolic space and the SPD manifold.

## 5.1 Hyperbolic Space

We perform numerous experiments in the hyperbolic setting. The purpose is twofold:

1. We wish to illustrate that our construction in Section 4 is not only more general, but also intrinsically more geometrically natural than pre-existing hyperbolic constructions such as HNN [17], and is thus able to learn better over hyperbolic data.

2. We would like to highlight that non-Euclidean learning benefits the most hyperbolic datasets. We can do this directly since our method provides a way to vary the geometry of a fixed neural network architecture, thereby allowing us to directly investigate the effect of changing geometry from Euclidean to hyperbolic.

### 5.1.1 Direct Comparison Against Hyperbolic Neural Networks [17]

To demonstrate the improvement of RResNet over HNN [17], we first perform node classification (NC) and link prediction (LP) tasks on graph datasets with low Gromov $\delta$-hyperbolicity [8], which means the underlying structure of the data is highly hyperbolic. The RResNet model is given the

|  | AFEW[13] | FPHA[18] | NTU RGB+D[48] | HDM05[39] |
|---|---|---|---|---|
| SPDNet | $33.24_{\pm 0.56}$ | $65.39_{\pm 1.48}$ | $41.47_{\pm 0.34}$ | $66.77_{\pm 0.92}$ |
| SPDNetBN | $35.39_{\pm 0.93}$ | $65.03_{\pm 1.35}$ | $41.92_{\pm 0.37}$ | $67.25_{\pm 0.44}$ |
| **RResNet Affine-Invariant** | $35.17_{\pm 1.78}$ | $\mathbf{66.53_{\pm 01.64}}$ | $41.00_{\pm 0.50}$ | $67.91_{\pm 1.27}$ |
| **RResNet Log-Euclidean** | $\mathbf{36.38_{\pm 1.29}}$ | $64.58_{\pm 0.98}$ | $\mathbf{42.99_{\pm 0.23}}$ | $\mathbf{69.80_{\pm 1.51}}$ |

Table 2: We run our SPD manifold RResNet on four SPD matrix datasets and compare against SPDNet [26] and SPDNet with batch norm [5]. We report the mean and standard deviation of validation accuracies over five trials and bold which method performs the best.

name "RResNet Horo." It utilizes a horosphere projection feature map-induced vector field described in Section 4. All model details are given in Appendix C.2. We find that because we adhere well to the geometry, we attain good performance on datasets with low Gromov $\delta$-hyperbolicities (e.g. $\delta = 0, \delta = 1$). As soon as the Gromov hyperbolicity increases considerably beyond that (e.g. $\delta = 3.5, \delta = 11$), performance begins to degrade since we are embedding non-hyperbolic data in an unnatural manifold geometry. Since we adhere to the manifold geometry more strongly than prior hyperbolic work, we see performance decay faster as Gromov hyperbolicity increases, as expected. In particular, we test on the very hyperbolic Disease ($\delta = 0$) [8] and Airport ($\delta = 1$) [8] datasets. We also test on the considerably less hyperbolic PubMed ($\delta = 3.5$) [47] and CoRA ($\delta = 11$) [46] datasets. We use all of the non-graph-based baselines from Chami et al. [8], since we wish to see how much we can learn strictly from a proper treatment of the embeddings (and no graph information). Table 1 summarizes the performance of "RResNet Horo" relative to these baselines.

Moreover, we find considerable benefit from the feature map-induced vector field over an embedded vector field that simply uses a Euclidean network to map from a manifold point embedded in $\mathbb{R}^n$. The horosphere projection captures geometry more accurately, and if we swap to an embedded vector field we see considerable accuracy drops on the two hardest hyperbolic tasks: Disease NC and Airport NC. In particular, for Disease NC the mean drops from 76.8 to 75.0, and for Airport NC we see a very large decrease from 96.9 to 83.0, indicating that geometry captured with a well-designed feature map is especially important. We conduct a more thorough vector field ablation study in Appendix C.5.

### 5.1.2 Impact of Geometry

A major strength of our method is that it allows one to investigate the direct effect of geometry in obtaining results, since the architecture can remain the same for various manifolds and geometries (as specified by the metric of a given Riemannian manifold). This is well-illustrated in the most hyperbolic Disease NC setting, where swapping out hyperbolic for Euclidean geometry in an RResNet induced by an embedded vector field decreases the F1 score from a 75.0 mean to a 67.3 mean and induces a large amount of numerical stability, since standard deviation increases from 5.0 to 21.0. We conduct a more thorough geometry ablation study in Appendix C.5.

### 5.2 SPD Manifold

A common application of SPD manifold-based models is learning over full-rank covariance matrices, which lie on the manifold of SPD matrices. We compare our RResNet to SPDNet [26] and SPDNet with batch norm [5] on four video classification datasets: AFEW [13], FPHA [18], NTU RGB+D [48], and HDM05 [39]. Results are given in Table 2. Please see Appendix C.6 for details on the experimental setup. For our RResNet design, we try two different metrics: the log-Euclidean metric [16] and the affine-invariant metric [12, 42], each of which captures the curvature of the SPD manifold differently. We find that adding a learned residual improves performance and training dynamics over existing neural networks on SPD manifolds with little effect on runtime. We experiment with several vector field designs, which we outline in Appendix B. The best vector field design (given in Section 4.2), also the one we use for all SPD experiments, necessitates eigenvalue computation. We note the cost of computing eigenvalues is not a detrimental feature of our approach since previous works (SPDNet [26], SPDNet with batchnorm [5]) already make use of eigenvalue computation[5]. Empirically, we observe that the beneficial effects of our RResNet construction are similar to those of the SPD batch norm introduced in Brooks et al. [5] (Table 2, Figure 5 in Appendix C.6). In addition, we find that our operations are stable with ill-conditioned input matrices, which commonly occur in the wild. To contrast, the batch norm computation in SPDNetBN, which relies on Karcher flow

---

[5]One needs this computation for operations such as the Riemannian $\exp$ and $\log$ over the SPD manifold.

| | Dataset Hyperbolicity | Disease $\delta = 0$ | Airport $\delta = 1$ | PubMed $\delta = 3.5$ | CoRA $\delta = 11$ |
|---|---|---|---|---|---|
| GNN | GCN [31] | $69.7_{\pm 0.4}$ | $81.4_{\pm 0.6}$ | $78.1_{\pm 0.2}$ | $81.3_{\pm 0.3}$ |
| | GAT [52] | $70.4_{\pm 0.4}$ | $81.5_{\pm 0.3}$ | $79.0_{\pm 0.3}$ | $\mathbf{83.0}_{\pm 0.7}$ |
| | SAGE [22] | $69.1_{\pm 0.6}$ | $82.1_{\pm 0.5}$ | $77.4_{\pm 2.2}$ | $77.9_{\pm 2.4}$ |
| | SGC [54] | $69.5_{\pm 0.2}$ | $80.6_{\pm 0.1}$ | $78.9_{\pm 0.0}$ | $81.0_{\pm 0.1}$ |
| GGNN | HGCN [8] | $74.5_{\pm 0.9}$ | $90.6_{\pm 0.2}$ | $\mathbf{80.3}_{\pm 0.3}$ | $79.9_{\pm 0.2}$ |
| | Fully HNN [10] | $\mathbf{96.0}_{\pm 1.0}$ | $90.9_{\pm 1.4}$ | $78.0_{\pm 1.0}$ | $80.2_{\pm 1.3}$ |
| | **G-RResNet Horo** | $95.4_{\pm 1.0}$ | $\mathbf{97.4}_{\pm 0.1}$ | $75.5_{\pm 0.8}$ | $64.4_{\pm 7.6}$ |

Table 3: Above we give node classification results for G-RResNet Horo compared with several graph-based neural network baselines (baseline methods and metrics are from Chami et al. [8]). Test F1 score is the metric reported. Mean and standard deviation are given over five trials. Note that G-RResNet Horo obtains a state-of-the-art result on Airport. As for the less hyperbolic datasets, G-RResNet Horo does worse on PubMed and does very poorly on CoRA, once more, as expected due to unsuitability of geometry. The GNN label stands for "Graph Neural Networks" and the GGNN label stands for "Geometric Graph Neural Networks."

[28, 35], suffers from numerical instability when the input matrices are nearly singular. Overall, we observe our RResNet with the affine-invariant metric outperforms existing work on FPHA, and our RResNet using the log-Euclidean metric outperforms existing work on AFEW, NTU RGB+D, and HDM05. Being able to directly interchange between two metrics while maintaining the same neural network design is an unique strength of our model.

## 6 Riemannian Residual Graph Neural Networks

Following the initial comparison to non-graph-based methods in Table 1, we introduce a simple graph-based method by modifying RResNet Horo above. We take the previous model and pre-multiply the feature map output by the underlying graph adjacency matrix $A$ in a manner akin to what happens with graph neural networks [54]. This is the simple modification that we introduce to the Riemannian ResNet to incorporate graph information; we call this method G-RResNet Horo. We compare directly against the graph-based methods in Chami et al. [8] as well as against Fully Hyperbolic Neural Networks [10] and give results in Table 3. We test primarily on node classification since we found that almost all LP tasks are too simple and solved by methods in Chami et al. [8] (i.e., test ROC is greater than $95\%$). We also tune the matrix power of $A$ for a given dataset; full architectural details are given in Appendix C.2. Although this method is simple, we see further improvement and in fact attain a state-of-the-art result for the Airport [8] dataset. Once more, as expected, we see a considerable performance drop for the much less hyperbolic datasets, PubMed and CoRA.

## 7 Conclusion

We propose a general construction of residual neural networks on Riemannian manifolds. Our approach is a natural geodesically-oriented generalization that can be applied more broadly than previous manifold-specific work. Our introduced neural network construction is the first that decouples geometry (i.e. the representation space expected for input to layers) from the architecture design (i.e. actual "wiring" of the layers). Moreover, we introduce a geometrically principled feature map-induced vector field design for the RResNet. We demonstrate that our methodology better captures underlying geometry than existing manifold-specific neural network constructions. On a variety of tasks such as node classification, link prediction, and covariance matrix classification, our method outperforms previous work. Finally, our RResNet's principled construction allows us to directly assess the effect of geometry on a task, with neural network architecture held constant. We illustrate this by directly comparing the performance of two Riemannian metrics on the manifold of SPD matrices. We hope others will use our work to better learn over data with nontrivial geometries in relevant fields, such as lattice quantum field theory, robotics, and computational chemistry.

**Limitations** We rely fundamentally on knowledge of geodesics of the underlying manifold. As such, we assume that a closed form (or more generally, easily computable, differentiable form) is given for the Riemannian exponential map as well as for the tangent spaces.

## Acknowledgements

We would like to thank Facebook AI for funding equipment that made this work possible. In addition, we thank the National Science Foundation for awarding Prof. Christopher De Sa a grant that helps fund this research effort (NSF IIS-2008102) and for supporting both Isay Katsman and Aaron Lou with graduate research fellowships. We would also like to acknowledge Prof. David Bindel for his useful insights on the numerics of SPD matrices.

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
