# Appendix

## A Riemannian Geometry: Relevant Reference Material

Here we give some relevant reference material that provides the reader with the fundamental operations used for the Poincaré ball model of hyperbolic space, as well as the two Riemannian SPD manifold structures employed.

### A.1 The Poincaré Ball Model

Hyperbolic space can be represented via several isometric models. We use the Poincaré ball model, which is defined by the set

$$\left\{ x \in \mathbb{R}^n \mid \|x\|_2^2 < -\frac{1}{K} \right\}, \tag{9}$$

where $K < 0$ is the space's constant negative curvature together with the metric given in the table below. We give a summary of hyperbolic operations in Table 4.

| Manifold | Euclidean $\mathbb{R}^n$ | Poincaré Ball $\mathbb{H}^n$ |
|---|---|---|
| Dimension, $\dim(\mathcal{M})$ | $n$ | $n$ |
| Metric $g_x$, | $g^{\mathbb{E}}$ | $(\lambda_x^K)g^{\mathbb{E}}$, where $g^{\mathbb{E}} = I$ |
| Tangent Space, $T_x\mathcal{M}$ | $\mathbb{R}^n$ | $\mathbb{R}^n$ |
| Projection, $\mathrm{proj}_{T_x\mathcal{M}}(v)$ | $v$ | $v$ |
| Exp Map, $\exp_x(v)$ | $x + v$ | $x \oplus_K \left( \tanh \left( \sqrt{\|K\|} \frac{\lambda_x^K \|v\|_2}{2} \right) \frac{v}{\sqrt{\|K\|}\|v\|_2} \right)$ |
| Geodesic Distance, $d(x,y)$ | $\|y - x\|_2$ | $\frac{1}{\sqrt{\|K\|}} \cosh^{-1} \left( 1 - \frac{2K\|x-y\|_2^2}{(1+K\|x\|_2^2)(1+K\|y\|_2^2)} \right)$ |

Table 4: Summary of Poincaré ball operations. We provide equivalent operations on Euclidean space for reference. $\oplus_K$ denotes Möbius addition [51], and $\lambda_x^K = \frac{2}{1+K\|x\|_2^2}$, a conformal factor.

### A.2 The SPD Manifold

We provide a summary of operations on the manifold of SPD matrices, in Table 5. For the SPD manifold, we illustrate the differences between the affine-invariant and log-Euclidean metrics. $\exp$ and $\log$ denote the matrix exponential and logarithm, respectively.

| Manifold | Euclidean $\mathbb{R}^n$ | $\mathbf{SPD}(n)$ Affine-Invariant | $\mathbf{SPD}(n)$ Log-Euclidean |
|---|---|---|---|
| Dimension, $\dim(\mathcal{M})$ | $n$ | $\frac{n(n+1)}{2}$ | $\frac{n(n+1)}{2}$ |
| Metric $g_x$, | $g^{\mathbb{E}}$ | $\mathrm{tr}(X^{-1}UX^{-1}V)$ | $\mathrm{tr}((D\log_X(U))^T D\log_X(V))$ |
| Tangent Space, $T_x\mathcal{M}$ | $\mathbb{R}^n$ | $\{V \mid V = V^T\}$ | $\{V \mid V = V^T\}$ |
| Projection, $\mathrm{proj}_{T_x\mathcal{M}}(v)$ | $v$ | $\frac{V+V^T}{2}$ | $\frac{V+V^T}{2}$ |
| Exp Map, $\exp_x(v)$ | $x + v$ | $X\exp(X^{-1}V)$ | $\exp(\log(X) + V)$ |
| Geodesic Distance, $d(x,y)$ | $\|y - x\|_2$ | $\|\log(X^{-1}Y)\|_F$ | $\|\log(Y) - \log(X)\|_F$ |

Table 5: Summary of SPD operations. We provide equivalent operations on Euclidean space for reference. We use both the affine-invariant and log-Euclidean metrics.

# B   Vector Field Design

Recall from the main paper that we can design a neural network-parameterized vector field $\ell_i : \mathcal{M} \to T\mathcal{M}$ for an embedded manifold $\mathcal{M}$ of dimension $D$, simply by defining a standard neural network $n_i : \mathbb{R}^D \to \mathbb{R}^D$ and then setting:

$$\ell_i(x) := \text{proj}_{T_x\mathcal{M}}(n_i(x)). \tag{10}$$

Though this vector field design is frequently trivial (assuming the manifold has a natural embedding in $\mathbb{R}^n$), it may be highly inefficient if an easy-to-implement but suboptimal embedding is used. This is especially the case if manifold structure is underexploited in the construction of such an embedding (see Section 4.1). In this section, we give a natural embedded vector field design for hyperbolic space, a more geometric feature map-induced vector field design for hyperbolic space, and explore a variety of possible vector field designs for the SPD manifold. In the general setting, note that obtaining a parsimonious (with respect to either representational dimension or parameter count) vector field design that is sufficiently expressive is nontrivial.

## B.1   Vector Field Design for Hyperbolic Space

For the embedded hyperbolic vector field design, we apply the general design construction referenced above. Note that $\mathbb{H}^n$ is an $n$-dimensional manifold with a trivial $\mathbb{R}^{n+1}$ embedding given by any coordinate representation. Thus we need only parameterize a neural network $n_i : \mathbb{R}^{n+1} \to \mathbb{R}^{n+1}$ and set

$$\ell_i(x) = \text{proj}_{T_x\mathbb{H}^n}(n_i(x)) \tag{11}$$

to obtain our neural network-parameterized vector fields. Observe that this vector field design is efficient and expressive, since $T_x\mathbb{H}^n \cong \mathbb{R}^n$, but is perhaps too expressive in that the vector field is not constructed around the geodesic geometry of hyperbolic space. For this, we employ the horosphere projection-induced vector field design introduced in Section 4.2 of the main paper. We simply fix a number of horospheres, randomly initialize them, and then further learn hyperparameters specifying a given horosphere.

## B.2   Vector Field Design for the SPD Manifold

Let $SPD(n)$ be the manifold of $n \times n$ SPD matrices with canonical metric, as in the main paper. We recall from Gallier and Quaintance [16] that $SPD$ has a Lie structure with algebra consisting of $n \times n$ symmetric matrices, denoted $S(n)$. The Riemannian exponential map (or equivalently, the matrix exponential map) is a bijection between $S(n)$ and $SPD(n)$. Recall by Lie symmetry [16] that the tangent space at $X \in SPD(n)$ is given by:

$$T_X SPD(n) = S(n) := \{P \mid P = P^T\}. \tag{12}$$

Observe that due to this tangent space structure, instead of utilizing the vector field construction given in Section 4.1 that requires an explicit projection operator, we may opt for more amenable designs oriented around the SPD manifold's Lie structure. We develop a variety of constructions below.

### B.2.1   Design 1: Embedded

We can observe that $SPD(n)$ is trivially embedded in $\mathbb{R}^{n^2}$, and so are its tangent vectors; we will use this observation to construct a simple vector field parameterization. Let $\text{vec} : \mathbb{R}^{n \times n} \to \mathbb{R}^{n^2}$ be row-major matrix vectorization and let $\text{vec}^{-1} : \mathbb{R}^{n^2} \to \mathbb{R}^{n \times n}$ be its inverse. Given a neural network $n_i : \mathbb{R}^{n^2} \to \mathbb{R}^{n^2}$ and an $X \in SPD(n)$, we may set:

$$\ell_i(X) = \text{proj}_{T_X SPD(n)}(\text{vec}^{-1}(n_i(\text{vec}(X)))) \tag{13}$$

where $\text{proj}_{T_X SPD(n)}$ symmetrizes a matrix in the tangent space of the identity matrix, before transforming it back to the tangent space of $X$. It is given by:

$$\text{proj}_{T_X SPD(n)}(V) = \frac{V + V^T}{2}. \tag{14}$$

Although this vector field representation is expressive, it also provides unneeded flexibility. For example, the intrinsic dimension of $T_X SPD(n) \cong S(n)$ is $\frac{n(n+1)}{2}$, but the $n_i$ map to all of $\mathbb{R}^{n^2}$. Based on this observation, we exploit tangent vector structure in the following vector field design to retain expressiveness while increasing efficiency.

### B.2.2 Design 2: Structured

Observe that our tangent spaces satisfy $T_X SPD(n) \cong S(n)$, and moreover that $SPD(n) \subset S(n)$. We know that $S(n)$ has dimension $\frac{n(n+1)}{2}$ since each symmetric matrix is uniquely determined by its upper triangular part. Let $\iota : \mathbb{R}^{\frac{n(n+1)}{2}} \hookrightarrow S(n)$ be the row-major injection of the upper triangular part into a symmetric matrix and let $\iota^{-1} : S(n) \twoheadrightarrow \mathbb{R}^{\frac{n(n+1)}{2}}$ be its inverse. Given a neural network $n_i : \mathbb{R}^{\frac{n(n+1)}{2}} \to \mathbb{R}^{\frac{n(n+1)}{2}}$ and an $X \in SPD(n)$, we may set:

$$\ell_i(X) = \iota(n_i(\iota^{-1}(X))). \tag{15}$$

Note that there is no longer any need for a projection to symmetric matrices, since we incorporate this structure directly into our vector field design. Moreover note that since $T_X SPD(n) \cong S(n) \cong \mathbb{R}^{\frac{n(n+1)}{2}}$, this vector field design is maximally expressive while being maximally efficient (representationally).

### B.2.3 Design 3: Parsimonious

Although Design 2 is maximally expressive and efficient, in some cases where expressivity is less of a concern we may want a a reasonable parsimonious vector field design. Our answer to this is to directly parameterize a symmetric matrix via its upper triangular portion. To be explicit, let our vector field be parameterized by euclidean parameters $v \in \mathbb{R}^{\frac{n(n+1)}{2}}$ and, for $X \in SPD(n)$, be given by:

$$\ell_i(X) = \iota(v) \tag{16}$$

This is a learnable vector field induced by a single tangent vector. Although highly efficient, its location-agnosticism makes it highly inexpressive.

### B.2.4 Design 4: Parsimonious Spectral

One may also consider exploiting manifold-specific structure in the context of Design 3 to produce a more expressive vector field that remains fairly efficient parametrically. A vector field design that accomplishes this is one that allows a map from the spectrum of the local SPD matrix to the spectrum of the symmetric matrix in the vector field construction. We let spec $: SPD(n) \to \mathbb{R}^n$ be the spectral map that takes SPD matrices to a vector of their eigenvalues, sorted in descending order. To be explicit, let our vector field be parameterized by $Q \in O(n)$[6], a neural network $f_i : \mathbb{R}^n \to \mathbb{R}^n$, and, for $X \in SPD(n)$, be given by:

$$\ell_i(X) = Q\mathrm{diag}(f_i(\mathrm{spec}(X)))Q^T \tag{17}$$

where diag $: \mathbb{R}^n \to \mathbb{R}^{n \times n}$ is the diagonal injection map. Observe that the spectrum of the symmetric matrix now depends locally on $X$, allowing for considerably more expressivity than in Design 3 at the cost of a low-dimensional neural network map $f_i : \mathbb{R}^n \to \mathbb{R}^n$. Moreover, the orthogonal constraint on $P$ may be preserved throughout optimization via one of a variety of easy-to-implement methods [1, 6].

Design 1 is naive, but very inefficient. Design 2 exploits manifold structure to be maximally efficient while being maximally expressive. Design 3 showcases the other extreme (relative to Design 1) and gives a maximally parsimonious vector field construction. Design 4 showcases a more flexible version of Design 3 that allows for considerably greater learning capability[7] while still being representationally efficient. The purpose of describing these designs is to underscore the trade-off between expressivity and parameter-efficiency in designing parameterized vector fields (Designs 1 and 2 vs. Designs 3 and 4) as well as the need to utilize manifold-specific structure to obtain a maximally expressive and efficient vector field design (Design 1 vs. Design 2). Additionally, we highlight that expressivity for parameter-constrained vector field designs can be nontrivially increased with insignificant overhead via the introduction of manifold-specific dependencies (Design 3 vs. Design 4).

---

[6]$O(n)$ is the group of orthogonal matrices.
[7]Verified empirically.

### B.3 Vector Field Design for Spherical Space

For the spherical vector field design, we again apply the general design construction referenced at the start of Appendix B. Similar to $\mathbb{H}^n$, $\mathbb{S}^n$ is an $n$-dimensional manifold which we treat as embedded in $\mathbb{R}^{n+1}$. Hence we parameterize a neural network $n_i : \mathbb{R}^{n+1} \to \mathbb{R}^{n+1}$ and set

$$\ell_i(x) = \text{proj}_{T_x \mathbb{S}^n}(n_i(x)) \tag{18}$$

to obtain our neural network-parameterized vector fields. As in the hyperbolic case, this vector field design is efficient and expressive, since $T_x \mathbb{S}^n \cong \mathbb{R}^n$.

### B.4 Feature Map-induced Vector Fields for General Manifolds

There is no perfect analog of a hyperplane for general manifolds. Hence, there is no immediately natural feature map in the general case. Despite this, we attempt to present a reasonable analog to hyperplane projection that extends to general manifolds. In particular, for a geodesically complete[8] manifold $\mathcal{M}$, consider specifying a pseudo-hyperplane by a point $p \in \mathcal{M}$ and a non-zero vector $v \in T_p\mathcal{M} \setminus \{\mathbf{0}\}$ whose orthogonal complement we exponentiate at the base point $p$ to give the following definition:

$$h_{p,v} = \exp_p(\{w \in T_p\mathcal{M} | w^T v = 0\}) \tag{19}$$

This definition[9] has the benefit of reducing to the usual Euclidean hyperplane definition when the manifold under consideration is $\mathbb{R}^n$. However, this hyperplane definition is not particularly suitable for general manifolds since it assumes geodesic completeness, which may not hold. Here we propose an alternative general definition of a hyperplane that exponentiates the intersection of a local orthogonal complement with a closed ball of radius $r$, $\bar{B}_r(0) \subset T_p\mathcal{M}$, given below:

$$h_{p,v,r} = \exp_p(\bar{B}_r(\mathbf{0}) \cap \{w \in T_p\mathcal{M} | w^T v = 0\}) \tag{20}$$

Notice that this $h_{p,v,r}$ pseudo-hyperplane is a strict generalization of $h_{p,v}$ that does not require geodesic completeness (since $r$ is finite), and that in the limit as $r \to \infty$ we recover $h_{p,v}$.

A general feature map can then be defined by projecting to such a pseudo-hyperplane:

$$g_{p,v,r}(x) = \min_{y \in h_{p,v,r}} d_{\mathcal{M}}(x, y) \tag{21}$$

where $d_{\mathcal{M}}$ is the induced geodesic distance on $\mathcal{M}$.

We test this general construction for hyperbolic space and compare it with the horosphere projection construction in Appendix C.4. The general construction performs reasonably well, but does not perform as well as the horosphere projection we give in this section. A more natural and performant manifold-dependent map can frequently be obtained by carefully considering the particular structure of the manifold (e.g. the spectral projection we give for $SPD(n)$).

## C Experimental Details

**Experiments on Hyperbolic Space**

### C.1 Datasets

We apply our hyperbolic RResNet to node classification and link prediction on four graph datasets with varying $\delta$-hyperbolicity.

**Airport ($\delta = 1$).** Airport is a dataset consisting of 2236 nodes where nodes represent airports and edges represent airline routes [8]. For node classification, each airport is given a label corresponding

---

[8]A manifold $\mathcal{M}$ is said to be geodesically complete if any geodesic can be followed indefinitely [32].

[9]This notion was originally introduced in the context of hyperbolic space in Ungar [51].

to the population of the country it is in. Each airport has a 4-dimensional feature vector consisting of geographic information.

**Pubmed** ($\delta = 3.5$) **and CoRA** ($\delta = 11$)**.** Pubmed and CoRA are both citation networks consisting of 2708 and 19717 nodes each [46, 47]. In citation networks, each node represents a paper and edges indicate a shared author between papers. Each node has a label consisting of what academic subareas the paper belongs to.

**Disease** ($\delta = 0$)**.** Disease is a synthetic dataset generated by simulating the SIR disease spreading model [8]. Node labels for classification indicate whether a node was infected or not and node features indicate a particular node's susceptibility to the disease.

## C.2 Architectural and Training Details

All of our testing uses the Poincaré ball model [41] to represent hyperbolic space. We use a similar setup to Chami et al. [8] to test RResNet's performance on hyperbolic space. First, in order to reduce the parameter count, we use a linear layer from the input dimension to a lower dimension before using RResNet as an encoder. For link-prediction tasks we use a Fermi-Dirac decoder and for node-classification tasks we use a linear decoder [8].

For our results using a feature map induced vector field, we take the projection onto a fixed number of horospheres. Each horosphere is randomly initialized with $\omega$ drawn uniformly from $\mathbb{S}^{n-1}$ and $b \sim \mathcal{N}(0, 1)$. We pass the horocycle projections to a linear layer followed by a Euclidean nonlinearity (typically ReLU [40]). During the training of each network, $\omega$ and $b$ are optimized using the same optimizer as the rest of the network. For further details regarding implementation, please see the accompanying Github code.

Horosphere projections are not the only natural feature map one can use, one alternative we experimented with was using parametetrized real eigenfunctions of the hyperbolic Laplacian as feature maps but we were unable to achieve similar performance to horosphere projections (results were significantly worse).

We use 250 horospheres for Disease, Airport, and CoRA and 50 horospheres for Pubmed. Models were trained for 500, 10000, 5000, and 5000 epochs for Disease, Airport, Pubmed, and CoRA, respectively, with the Adam optimizer [30]. All other hyperparameters, such as learning rate and weight decay, were determined using random search.

All experiments were run on a single NVIDIA Quadro RTX A6000 48GB GPU.

## C.3 Comparison Between Embedded and Horocycle-induced Vector Field Designs

| Dataset | Disease ($\delta = 0$) | Airport ($\delta = 1$) | Pubmed ($\delta = 3.5$) | CoRA ($\delta = 11$) |
|---|---|---|---|---|
| **RResNet Embedded** | $75.0_{\pm 5}$ | $83.0_{\pm 2.0}$ | $\mathbf{73.2}_{\pm 1.0}$ | $\mathbf{59.6}_{\pm 1.0}$ |
| **RResNet Horo** | $\mathbf{76.8}_{\pm 2.0}$ | $\mathbf{96.9}_{\pm 0.3}$ | $71.4_{\pm 2.2}$ | $52.4_{\pm 5.5}$ |

Table 6: Node classification results for RResNet with two different vector field designs (test F1 score is the metric given).

In order to investigate the effect vector field design has, we look at the performance of RResNet when using the embedded or horosphere projection-induced vector field in Table 8. On more hyperbolic datasets (Disease and Airport), the more geometrically principled design attains higher F1 scores. This effect is reversed on the less hyperbolic datasets (Pubmed and CoRA), indicating that a more geometrically principled vector field only helps when the data geometry is similar to the model geometry, as expected.

## C.4 Comparison Between Horocycle-induced and Pseudo-Hyperplane-induced Vector Field Designs

In Table 7 we compare the RResNet construction with vector fields induced by projection to pseudo-hyperplanes (as defined in the main paper in Section 4.2) for hyperbolic space (RResNet Pseudo-Hyperplane) to the horocycle projection-induced vector field RResNet construction (RResNet Horocy-

| Dataset | Disease ($\delta = 0$) | Airport ($\delta = 1$) | Pubmed ($\delta = 3.5$) | CoRA ($\delta = 11$) |
|---|---|---|---|---|
| **RResNet Horocycle** | **76.8**$_{\pm 2.0}$ | **96.9**$_{\pm 0.3}$ | **71.4**$_{\pm 2.2}$ | **52.4**$_{\pm 5.5}$ |
| **RResNet Pseudo-Hyperplane** | **77.2**$_{\pm 0.4}$ | 90.3$_{\pm 4.5}$ | 66.7$_{\pm 5.0}$ | 41.4$_{\pm 5.7}$ |

Table 7: Node classification results for RResNet with two different vector field designs (test F1 score is the metric given).

cle). Note that RResNet Pseudo-Hyperplane performs worse for most tasks, although the construction is more general (as mentioned in the main paper).

## C.5 Ablation Study

**Nonlinearity Ablation**

| Dataset | Disease ($\delta = 0$) | Airport ($\delta = 1$) | Pubmed ($\delta = 3.5$) | CoRA ($\delta = 11$) |
|---|---|---|---|---|
| **RResNet Horo w/o Nonlinearity** | 71.9$_{\pm 9.2}$ | **96.9**$_{\pm 3.0}$ | **71.2**$_{\pm 1.1}$ | 49.6$_{\pm 2.0}$ |
| **RResNet Horo** | **76.8**$_{\pm 2.0}$ | **96.9**$_{\pm 0.3}$ | **71.4**$_{\pm 2.2}$ | **52.4**$_{\pm 5.5}$ |

Table 8: Node classification results for RResNet with and without a nonlinearity between layers (test F1 score is the metric given).

To study the expressiveness of the horocycle induced vector field design, we ablate the nonlinearity in the vector field. With the nonlinearity, the F1 score either increases or remains the same across all datasets, which the advantage being most pronounced for Disease.

**Geometry Ablation**

| Dataset | Disease ($\delta = 0$) |
|---|---|
| RResNet Embedded (Euclidean) | 67.3$_{\pm 21.0}$ |
| **RResNet Embedded (Hyperbolic)** | 75.0$_{\pm 5.0}$ |
| RResNet Feature Map (Euclidean) | 73.1$_{\pm 3.4}$ |
| **RResNet Feature Map (Hyperbolic)** | **76.8**$_{\pm 2.0}$ |

Table 9: Node classification results of RResNet with different vector field designs and model geometry (test F1 score is the metric given). When swapping geometry for a specific model, all hyperparameters are kept the same, which we are able to do easily with our architecture.

We look at the performance of varying RResNets on the most hyperbolic dataset to identify the effect model geometry has in Table 9. As expected, using hyperbolic space yields higher F1 scores with lower standard deviations. In particular, the high standard deviation of 21.0 for "RResNet Embedded (Euclidean)" indicates that it fails to properly learn in a number of trials.

**Residual Connection Ablation**

It is reasonable to try other residual connection implementations outside of our natural geometric Riemannian exp-map based implementation. In particular, one may try to implement a Riemannian residual neural networks directly via a gyrovector [51] addition. We give the results in Table 10 and show that not only is this method less desirable geometrically, but also gives worse results on our chosen benchmarks. The Euclidean model is given as a baseline and the Riemannian ResNet here is a reference. All models are implemented with a comparable number of parameters and are two layer residual neural networks.

| Dataset | Airport ($\delta = 1$) |
|---|---|
| **Euclidean** | $69.4_{\pm 1.8}$ |
| **Gyrovector** | $60.8_{\pm 0.9}$ |
| **RResNet Horo** | $\mathbf{75.9}_{\pm 2.5}$ |

Table 10: Node classification results for RResNet with three different residual connection designs (test F1 score is the metric given).

**Experiments on the SPD Manifold**

## C.6 Datasets

We apply our SPD architecture on four different video recognition tasks. For all tasks, we generate covariance or correlation matrices sampled from each video's frames. Given frames $t \in \{1, \dots, T\}$ and their corresponding feature vectors $x_t \in \mathbb{R}^n$, we generate a $n \times n$ covariance matrix by sampling the frames: $\mathrm{X} = \frac{1}{T-1} \sum_{t=1}^{T} (x_t - \mu)(x_t - \mu)^T$. Optionally, we can divide the matrices by the standard deviations to instead generate correlation matrices. For certain tasks, we find that these have better conditioning.

While covariance and correlation matrices are positive semi-definite, they are not necessarily SPD. In fact, they are only SPD if the set of sampled vectors, $\{x_1, \dots, x_T\}$, consists of $n$ linearly-independent vectors. If the sampled vectors $x_t, x_{t+1}$, are similar, which is the case for neighboring frames of a video, the matrices may be close to singular. This phenomenon poses issues in downstream tasks such as taking a matrix logarithm, which can create numerical instability. For all tasks, we preprocess our data by removing covariance matrices which fail a Cholesky decomposition.

**AFEW.** AFEW [13] is an emotion recognition dataset consisting of 1,345 videos and 7 classes. As done in Brooks et al. [5], Huang and Gool [26], we use covariance matrices created from $20 \times 20$ video frames, flattened into 400-dimensional $x_t$ vectors.

**FPHA.** The First-Person Hand Action Benchmark (FPHA) [18] consists of 1,175 videos of humans performing 45 different tasks. The dataset includes the $(x, y, z)$ coordinates of 21 joint locations from a human hand. Following the approach of Hussein et al. [27], for each frame, we flatten the coordinates into a 63-dimensional vector $x_t$. We then take the correlation matrices. We use subjects 1-3 for training and 4-6 for validation.

**NTU RGB+D.** NTU RGB+D [48] is an action recognition dataset which includes the 3D locations of 25 body joints. NTU RGB+D is a large scale dataset with 56,880 videos and 60 tasks. For our $x_t$ vectors, we use the flattened versions of 3D joint coordinates as feature vectors. Our matrices have dimension 75.

**HDM05.** Mocap Database HDM05 [39] is another action recognition dataset which includes 3D locations of 31 joints. Following the task designed in Huang and Gool [26], the goal is to classify each video clip into one of 117 action classes. We use the covariance matrices provided in Brooks et al. [5].

## C.7 Architectural Details

Given a dataset of covariance matrices, our goal is to classify a matrix into one of several classes. To illustrate, we give our architecture for the AFEW task as an example. Because of how costly it would be to parameterize vector fields at this dimension, we use a BiMap layer [26], $\mathrm{BiMap}_{d_{i+1}}^{d_i} : SPD(d_i) \to SPD(d_{i+1})$ as a base point remapping from $400 \times 400$ matrices to $50 \times 50$ matrices. We use vector field design 4 from Appendix B. In the context of this problem, we have:

$$\ell_1(X) = Q\mathrm{diag}(f_1(\mathrm{spec}(X)))Q^T \tag{22}$$

where $f_1 : \mathbb{R}^{50} \to \mathbb{R}^{50}$, $\mathrm{spec} : SPD(50) \to \mathbb{R}^{50}$, $P \in O(50)$ (spec is defined above in Appendix B). In practice, we experiment with a variety of $f_1$ designs, such as sequences of linear layers or 1D convolutions. Note the vector field is a map $\ell_1 : SPD(50) \to TSPD(50)$. We express our forward pass as

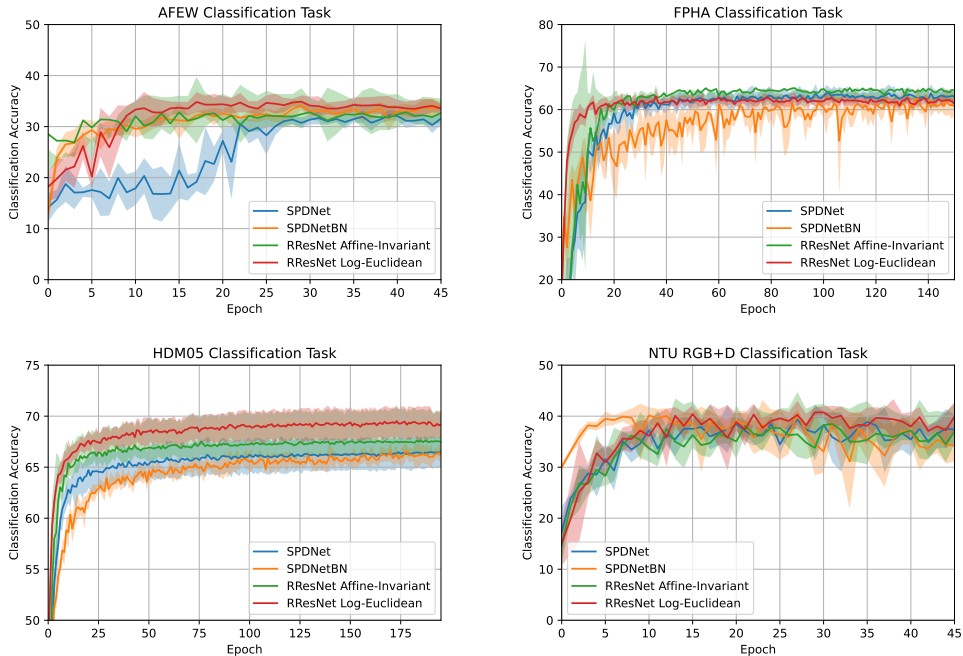

Figure 5: Validation accuracies for our RResNet compared to the SPDNet [26] and SPDNetBN [5] baselines. For each model, results are averaged over five trials. Error bars denote one standard deviation away from the mean accuracy. We observe that our model converges faster and achieves higher accuracies than SPDNet and SPDNetBN.

$$g(x) = \exp_{\mathbf{BiMap}_{50}^{400}(x)}(\ell_1(\mathbf{BiMap}_{50}^{400}(x))) \tag{23}$$

which is a map $g : SPD(400) \to SPD(50)$. Our $\exp$ map depends on the Riemannian metric we choose on the manifold. Thereafter we apply a logarithm to the eigenvalues of the $50 \times 50$ matrices (this helps linearize features [5]). Lastly we flatten the matrices and use a linear map from dimension 2500 to dimension 7 (representing the 7 different emotions). We use a simple cross entropy loss [19] to train the model.

## C.8 Results

We compare our RResNet design above (Appendix C.7) to SPDNet [5, 26], a network architecture for SPD matrix learning. All models have a comparable number of parameters. To replicate the results of Brooks et al. [5], we use a learning rate of $5 \cdot 10^{-2}$ for the baseline. We find that any higher learning rate causes training instability. However, we observe that our model remains stable with a learning rate of $1 \cdot 10^{-1}$. Our model has faster convergence and achieves a higher accuracy than SPDNet and SPDNetBN (see Table 2 in the main paper and Figure 5 above). Moreover, our model's ability to switch out geometries (as given by the log-Euclidean and affine-invariant metrics) gives the ability to outperform prior work on all tasks.

## C.9 Comparison Between Vector Field Designs

For the SPD manifold, we illustrate differences between our four different vector field designs outlined in Appendix B on the AFEW task. Results are given in Table 11. Note that the chosen spectral map-induced vector field is very efficient in terms of parameter count and performs best in terms of accuracy.

## C.10 Ablation Study

**Nonlinearity Ablation**

|  | AFEW[13] | Number of Parameters |
|---|---|---|
| Naive | $34.90_{\pm 0.74}$ | 6,290,007 |
| Structured | $34.76_{\pm 1.07}$ | 1,664,407 |
| Parsimonious | $34.82_{\pm 1.82}$ | 38,782 |
| **Spectral Map** | $\mathbf{36.38_{\pm 1.29}}$ | 45,057 |

Table 11: We compare the accuracy of our four vector field designs for the SPD manifold. We see that the spectral map provides the best balance of accuracy and parameter efficiency.

We ablate the nonlinearity in the spectral map design in Table 12, and find that the nonlinearity slightly improves performance. For AFEW, we use a one layer vector field, which is why the reported accuracies are the same.

|  | AFEW | FPHA | NTU RGB+D | HDM05 |
|---|---|---|---|---|
| Aff-Inv w/o Nonlinearity | $35.17_{\pm 1.78}$ | $65.03_{\pm 2.22}$ | $41.27_{\pm 0.22}$ | $65.92_{\pm 1.27}$ |
| Aff-Inv | $35.17_{\pm 1.78}$ | $\mathbf{66.53_{\pm 1.64}}$ | $41.00_{\pm 0.50}$ | $67.91_{\pm 0.66}$ |
| Log-Euc w/o Nonlinearity | $36.38_{\pm 1.29}$ | $65.25_{\pm 2.14}$ | $42.87_{\pm 0.83}$ | $68.73_{\pm 1.75}$ |
| Log-Euc | $\mathbf{36.38_{\pm 1.29}}$ | $64.58_{\pm 0.98}$ | $\mathbf{42.99_{\pm 0.23}}$ | $\mathbf{69.80_{\pm 1.51}}$ |

Table 12: We show that classification accuracy either improves or remains the same with the nonlinearity. For AFEW, we use one layer in the vector field, which is why the reported accuracies are the same.

### Geometry Ablation

We study the geometry of the SPD manifold by comparing our Riemannian ResNet to a Euclidean ResNet. For the Euclidean network, we treat each matrix as a Euclidean vector by flattening it into a length $n \times n$ vector. We then pass it through a Euclidean ResNet. Our results in Table 13 show that the Riemannian ResNets (Aff-Inv and Log-Euc) perform significantly better across all datasets.

|  | AFEW | FPHA | NTU RGB+D | HDM05 |
|---|---|---|---|---|
| Euclidean | $30.08_{\pm 1.36}$ | $30.72_{\pm 1.03}$ | $34.63_{\pm 3.10}$ | $0.80_{\pm 0.10}$ |
| Aff-Inv | $35.17_{\pm 1.78}$ | $\mathbf{66.53_{\pm 1.64}}$ | $41.00_{\pm 0.50}$ | $67.91_{\pm 1.27}$ |
| Log-Euc | $\mathbf{36.38_{\pm 0.24}}$ | $64.58_{\pm 0.98}$ | $\mathbf{42.99_{\pm 0.23}}$ | $\mathbf{69.80_{\pm 1.51}}$ |

Table 13: We show that the Euclidean ResNet performs worse across all datasets, and fails for HDM05.

### Residual Connection Ablation

Similar to the gyrocalculus used in Ganea et al. [17], López et al. [34] have extended gyrovector operations to the manifold of SPD matrices. In particular, the authors define Möbius addition as $X \oplus Y = \sqrt{X}Y\sqrt{X}$ for SPD matrices $X, Y$. It is reasonable to ask how this purely algebraic, non-geometric construct performs when used to implement a residual connection. With this choice of addition, the residual connection for a ResNet specific to the SPD manifold would have the form $\ell_i(X) + X = \sqrt{\ell_i(X)}X\sqrt{\ell_i(X)}$. In Table 14, we show that this choice of addition struggles to reach the accuracy of our Riemannian ResNet design.

### Experiments on Spherical Space

### C.11   Dataset

We wish to explore the generality of our method: in particular, our ability to vary geometry without constructing entirely new operations for each manifold. We repeat one of the experiments tested on our hyperbolic RResNet, swapping out the hyperbolic manifold for the spherical manifold.

|            | AFEW              | FPHA              | NTU RGB+D         | HDM05             |
|------------|-------------------|-------------------|-------------------|-------------------|
| Gyrovector | $23.23_{\pm 0.98}$ | $61.33_{\pm 4.74}$ | $40.77_{\pm 3.10}$ | $5.69_{\pm 2.15}$ |
| Aff-Inv    | $35.17_{\pm 1.78}$ | $\mathbf{66.53}_{\pm 1.64}$ | $41.00_{\pm 0.50}$ | $67.91_{\pm 1.27}$ |
| Log-Euc    | $\mathbf{36.38}_{\pm 1.29}$ | $64.58_{\pm 0.98}$ | $\mathbf{42.99}_{\pm 0.23}$ | $\mathbf{69.80}_{\pm 1.51}$ |

Table 14: We show that the Riemannian ResNet model with Möbius addition struggles to reach the classification accuracies of our exponential map design. The difference is most pronounced on HDM05, where the gyrovector model struggles to learn meaningful representations.

**CoRA.** This dataset is described above in Appendix C.1. With $\delta = 11$, CoRA is the least hyperbolic of the datasets tested with our hyperbolic RResNet. As such, we wanted to try swapping the RResNet geometry to better match the data geometry.

## C.12   Architectural Details

The design of our spherical RResNet is identical to that of our hyperbolic RResNet (described in Appendix C.2), aside from switching the geometric representation from hyperbolic to spherical. As before, we first have a linear layer to move from the input dimension to a lower dimension. Then we use our RResNet as an encoder. Here we only test link prediction, so we use a Fermi-Dirac decoder.

We train for 2000 epochs using the Adam optimizer [30], and we again found all hyperparameters via random search.

## C.13   Results

We give results for link prediction on CoRA, displayed in Table 15. Mean and standard deviation across 5 separate trials are reported.

|        | **Dataset** **Hyperbolicity** | CoRA $\delta = 11$ |
|--------|-------------------|-------------------|
| Hyp    | RResNet           | $88.9_{\pm 0.2}$  |
|        | RResNet Graph     | $87.6_{\pm 0.9}$  |
| Sphere | RResNet           | $90.7_{\pm 1.0}$  |
|        | RResNet Graph     | $\mathbf{91.7}_{\pm 0.4}$ |

Table 15: Test accuracy of various models, in terms of ROC AUC.

We find that even the most basic spherical RResNet design, which does not use a feature map, outperforms both hyperbolic RResNets. This indicates that our model improves when endowed with geometry more suitable for given data. Additionally, our model's flexibility allows us to easily obtain such results without altering the architecture.

# D   Theoretical Results

In this section we give a variety of theoretical results that demonstrate the principled nature of our Riemannian ResNet construction.

## D.1   Reduction to Standard ResNet in Euclidean Case

We show that our construction agrees with the standard ResNet when the underlying manifold is Euclidean space and when we are using the embedded vector field design. This aligns with our intuition and shows that our construction is a natural generalization of previous work.

**Proposition 1.** *When $\mathcal{M}^{(i)} \cong \mathbb{R}^{d_i}$, our RResNet with the embedded vector field design is a standard residual network.*

*Proof.* Note that the embedded vector fields take the form:
$$\ell_i(x) = \text{proj}_{T_x \mathbb{R}^n}(n_i(x)) = n_i(x) \tag{24}$$

for a parameterized neural network $n_i : \mathbb{R}^{d_{i-1}} \to \mathbb{R}^{d_i}$, meaning that our $\ell_i$ are standard neural networks. The $h_i : \mathbb{R}^{d_{i-1}} \to \mathbb{R}^{d_i}$ can be replaced by Euclidean linear layers that go from dimension $d_{i-1}$ to dimension $d_i$. Also observe since $\exp_x(v) = x + v$, our neural network construction becomes:

$$f(x) = x^{(m)} \tag{25}$$

$$x^{(0)} = x \tag{26}$$

$$x^{(i)} = \exp_{h_i(x^{(i-1)})}(\ell_i(h_i(x^{(i-1)}))) \tag{27}$$

$$= h_i(x^{(i-1)}) + \ell_i(h_i(x^{(i-1)})) \tag{28}$$

$$= h_i(x^{(i-1)}) + n_i(h_i(x^{(i-1)})) \tag{29}$$

where the last equality holds $\forall i \in [m]$. Moreover, if all $d_i$ are the same, we can use the identity map for our $h_i$, implying:

$$x^{(i)} = x^{(i-1)} + n_i(x^{(i-1)}) \; \forall i \in [m] \tag{30}$$

Hence our neural network architecture reduces precisely to that of Euclidean residual neural networks.
$\square$

## D.2 Hyperbolic Neural Networks (HNNs) [17] Learn via a Hyperbolic Bias

We make note of the fact that although the gyrovector generalization of Euclidean networks offered by Ganea et al. [17] is algebraic and generalizable to many manifolds such as hyperbolic space and the manifold of SPD matrices [34], the linear layer of the construction is Euclidean, except for the hyperbolic bias addition. We illustrate this in what follows.

**Proposition 2.** *For $x \in \mathbb{H}^n$ and hyperbolic matrix-vector multiplication [17] defined by*

$$M^{\otimes}(x) = \tanh\left(\frac{||Mx||}{||x||} \tanh^{-1}(||x||)\right) \frac{Mx}{||Mx||} \tag{31}$$

*where $M : \mathbb{R}^n \to \mathbb{R}^n$ is a linear map, we have*

$$M_2^{\otimes}(M_1^{\otimes}(x)) = \tanh\left(\frac{||M_2||||M_1x||}{||x||} \tanh^{-1}(||x||)\right) \frac{M_2M_1x}{||M_2||||M_1x||} = (M_2M_1)^{\otimes}(x) \tag{32}$$

*Proof.* For two linear maps of the same size $M_1, M_2$ we have:

$$M_2^{\otimes}(M_1^{\otimes}(x)) = \tanh\left(\frac{||M_2M_1^{\otimes}(x)||}{||M_1^{\otimes}(x)||} \tanh^{-1}(||M_1^{\otimes}(x)||)\right) \frac{M_2M_1^{\otimes}(x)}{||M_2M_1^{\otimes}(x)||} \tag{33}$$

$$= \tanh\left(\frac{||M_2M_1^{\otimes}(x)||}{\tanh\left(\frac{||M_1x||}{||x||}\tanh^{-1}(||x||)\right)} \left(\frac{||M_1x||}{||x||}\tanh^{-1}(||x||)\right)\right) \frac{M_2M_1^{\otimes}(x)}{||M_2M_1^{\otimes}(x)||} \tag{34}$$

$$= \tanh\left(\frac{||M_2||||M_1x||}{||x||}\tanh^{-1}(||x||)\right)\frac{M_2M_1x}{||M_2||||M_1x||} = (M_2M_1)^{\otimes}(x) \tag{35}$$
$\square$

We see that we have cancellation that de facto reduces the learning of two hyperbolic linear layers with no hyperbolic bias to the learning of a single hyperbolic layer. Inductively, this precise argument applies to any number of layers. This reduction is characteristic to what one sees in the case of Euclidean networks, and more importantly, from the above equation we see that learning hyperbolic linear layers de facto reduces to learning Euclidean linear maps ($M_1$ and $M_2$ above) that are placed in between an initial Riemannian $\log$ map (taken at the origin) and a trailing Riemannian $\exp$ map (taken at the origin).

Thus, the main non-Euclidean, hyperbolic construct in Ganea et al. [17] is the hyperbolic bias, introduced in Section 3.2 of Ganea et al. [17]. Our method is distinctly different in that even simple residual linear layers make use of geodesic information; hence, learning does not reduce to the Euclidean case.

# E   Comparison with Other Constructions

Here we elaborate on how our method compares with other constructions, elucidating a claim made in the main paper. We note that compared to other methods, our construction is fully general (in the sense that it extends to all Riemannian manifolds) and better conforms with geometry. For example, general methods like HNN [17], HGCN [8], and SPDNetBN [5] use the fact that hyperbolic space and the SPD manifold are spaces with everywhere non-negative curvature, meaning that geodesics are unique. As such, core building blocks of these models globally project to a Euclidean space via a map known as the Riemannian $\log$ map, which can be thought of as an inverse to the exponential map. This global projection relies on the choice of a particular base point (equivalent to the base point in the $\exp$ map definition), which is arbitrarily selected. Once this projection has taken place, prior methods usually simply perform a Euclidean operation, and project back to the manifold via the usual Riemannian $\exp$. This system does not generalize to manifolds which are not globally diffeomorphic to Euclidean space (note this is quite restrictive and different from being locally diffeomorphic to Euclidean space), and furthermore, the reliance on fundamentally Euclidean operations and arbitrary base point destroys the geodesic geometry: the log map can be thought of as linearizing manifold geometry with respect to a certain base point—the further away points are from this base point, the more distorted the projected geometry.

More specialized methods like Chen et al. [10], Shimizu et al. [49] instead work with algebraic operations that exist only for hyperbolic space. These do not generalize to arbitrary manifolds, which limits potential applications. By comparison, our method simultaneously generalizes to arbitrary manifolds and directly conforms with underlying geometry.