# OpenReview forum: "Riemannian Residual Neural Networks"
_NeurIPS.cc/2023/Conference — NeurIPS 2023 poster_

### Official Review · Reviewer_nyBA · 2023-07-01

**Soundness:** 2 fair
**Presentation:** 1 poor
**Contribution:** 2 fair
**Rating:** 4
**Confidence:** 3

**Summary:**

EDIT: Having read everything here. I am increasing my score slightly. However, I still think the paper is not clearly explained. It is unclear why you first introduce the method without the feature map. It seems like there are two different versions of the method and it is not clear which is which. I am also confused by the experimental set up for datasets such as CORA. Do you first embed the graphs into a manifold?

Additionally, I think that requiring a closed form expression for geodesics is somewhat strong and limits the applicability.


The authors claim to attempt RESNET structures to hyperbolic manifolds

**Strengths:**

The paper is applicable to a wide variety of manifolds

**Weaknesses:**

The proposed methods seem to not really leverage the manifolds intrinsice geometry and depend entirely on the embedding on the manifold into ambient space $R^D$.

Indeed $n_i$ is defined on all of $R^D$. This means that there is no gauranteed that there would be any notion of consistence if $M$ was embedded into $R^D$ and $R^{D'}$ two different ways (where $D'$ may or may not equal $D$). This seems to be a major limitation of this that is not properly discussed. at a bare minumum, there should be some level of invariance to, e.g., Rotations and Translations of the manifold in $R^D$ after its been embedded.

Additionally, much of the paper is hard to understand such as the construction of the feature maps, which appears to take place in local coordinate systems which will not be consistent across the manifold.

**Questions:**

N/A

---

> ### Author Rebuttal · Authors · 2023-08-08
>
> > The proposed methods seem to not really leverage the manifolds intrinsice geometry and depend entirely on the embedding on the manifold into ambient space $R^D$
>
> A: Our approach uses intrinsic geometry (we use local coordinates, as mentioned by the reviewer).
>
> ---
>
> > Indeed $n_i$ is defined on all of $R^D$. This means that there is no gauranteed that there would be any notion of consistence if $M$ was embedded into $R^D$ and $R^{D’}$ two different ways (where $D’$ may or may not equal $D$). This seems to be a major limitation of this that is not properly discussed. at a bare minumum, there should be some level of invariance to, e.g., Rotations and Translations of the manifold in $R^D$ after its been embedded.
>
> A: $n_i$ is not invariant, but this is a necessary cost to realize our manifolds for computation and is standard practice in the literature [36, 38]. However, our feature map-based layers are notably invariant to embedding, which was the motivation for introducing them.
>
> Our feature maps are invariant to local coordinates and are consistent (e.g. SPD eigenvalues are invariant to conjugation and the hyperbolic space construction is invariant to choice of model).

---

### Official Review · Reviewer_5HC1 · 2023-07-02

**Soundness:** 3 good
**Presentation:** 3 good
**Contribution:** 3 good
**Rating:** 6
**Confidence:** 4

**Summary:**

This paper extends the well known residual network which are usually applied to Euclidean data to a variant defined on manifold. The main novelty is to replace the "addition / plus" operation in Euclidean space to exponential operation. Specifically, given an input point on a certain manifold (e.g., hyperbolic or SPD), it first learns a vector in the tangent space of the given input point through a neural network layer, and then maps the learned vector back to the manifold using exponential. By utilizing pushforward and pullback operations, the proposed method can transform input between different manifolds with different dimensions. Experiments are conducted on hyperbolic and SPD spaces to demonstrate the superior performance of the proposed method over HNN and SPDNet.

**Strengths:**

This paper proposed to extend residual network from Euclidean space to non-linear manifold by replacing the conventional addition/plus operation with manifold exponential operation. The theoretical part is sound and the experiments are effective in supporting the proposed method.

**Weaknesses:**

The biggest weakness, as also mentioned in by the authors at the end of the paper, is whether the proposed method is only applicable to hyperbolic and SPD matrix? Is it possible to apply this residual network to other non-linear manifolds that have closed-form exponential manifolds? Furthermore, Is it possible to apply this residual network to other non-linear manifolds that do $\textbf{not}$ have closed-form exponential manifolds? If so, please list such manifolds.

**Questions:**

In general:

1. Except for Euclidean, hyperbolic and SPD, any other manifold are applicable to the proposed residual network?

Method part:

2. For the function h_i defined in Line 202 parameterized by a neural network, given an input on a mainfold M_{i-1}, how to ensure the output is also on a manifold M_i?

3. For SPD matrix, what's the advantage of the proposed Riemannian Residual Neural Networks over SPDNet (the AAAI paper) in theory?

4. In Line 225 - 231, what's the relationship between f_i and g_{\theta_i}?

5. Still in Line 225 - 231, why does $\nabla g_{\theta_i}$ is a map from M to TM?

6. In SPD case of Line 258 - 260, since g_k does not contain any learnable parameters, how to train such a network for SPD?

Experiment part:

7. How many links on average per graph used in the experiment in Section 5.1.1? It is important to verify the proposed method could perform well on medium-to-large datasets.

8. In Table 2, the standard deviations of the proposed method is obviously larger than SPDNet, does it imply the proposed method is unstable than SPDNet? (Similar phenomena also happen in Table 1, though not that obvious.)

Implementation part:

9. As some parts (e.g., Line 225 - 231) are not that obvious to implement, will the code be released?



**Limitations:**

The authors mentioned the limitations of the proposed method in Line 357 - 359.

---

> ### Author Rebuttal · Authors · 2023-08-08
>
> Thank you for the review and the constructive comments. We appreciate that you think our idea is theoretically sound and our experiments are supportive. We address your comments from the “Weaknesses” and "Questions" sections below.
>
> ---
>
> > The biggest weakness... closed-form exponential maps?
>
> A: Riemannian ResNets can be applied to more than just hyperbolic space and the manifold of SPD matrices; the answer to both questions posed is an unequivocal “yes.” So long as a closed-form exp map is provided, our method becomes immediately applicable. In fact, we include experiments on spherical space (which does have a closed form exponential map) in sections C.11 through C.13 of the appendix. We provide a general feature map construction in B.5 which is applicable to any Riemannian manifold. Please see the question below for an explicit list of some of these manifolds.
>
> ---
>
> > Except for Euclidean, hyperbolic and SPD, any other manifold are applicable to the proposed residual network?
>
> A: Yes, as mentioned above our method is easily applicable to myriad manifolds assuming there is a closed form exponential map provided. Common cases outside of the Euclidean, hyperbolic, and SPD manifolds include the manifold of spherical space and the Grassmannian manifold. Every matrix Lie group is also included, so for example the manifold of special unitary matrices (SU(n)), the manifold of special orthogonal matrices ($SO(n)$), and the manifold of all invertible matrices ($GL(n, \mathbb{R})$).
>
> ---
>
> > Is it possible to apply this residual network to other non-linear manifolds that do not have closed-form exponential manifolds?
>
> A: We note that without a closed-form exponential map, we would need a differentiable ODE solver for the geodesic equations, immediately complicating the computation by a significant amount. However, we note that this is not a considerable limitation of our method in that nearly all prior work in this subfield requires at least a closed-form exponential map, see for examples Neural Manifold ODEs [36], Riemannian Continuous Normalizing Flows [38], and Riemannian Convex Potential Maps [https://arxiv.org/abs/2106.10272].
>
> ---
>
> > For the function $h_i$ defined in Line 202 parameterized by a neural network, given an input on a manifold $M_{i-1}$, how to ensure the output is also on a manifold $M_i$?
>
> A: We assume in our construction that $h_i$ maps $M^{(i-1)}$ to $M^{(i)}$. In general, one has to carefully construct these maps to ensure the output stays on manifold. As a concrete example, our h_i for the SPD case map from an SPD matrix of one dimension to another by conjugating with a Stiefel matrix [26].
>
> ---
>
> > For SPD matrix, what's the advantage of the proposed Riemannian Residual Neural Networks over SPDNet (the AAAI paper) in theory?
>
> A: Besides the empirical benefits over SPDNet, theoretically, our work also generalizes SPDNet. SPDNet only operates on matrices under the log-Euclidean metric, which endows the manifold with flat geometry (the sectional curvature is zero at every point). This model fails to provide a way to capture more nontrivial geometry of the SPD manifold, a drawback that our approach removes.
>
> We demonstrate that the proposed Riemannian ResNet can learn over the log-Euclidean metric, and the affine-invariant metric, which has non-constant sectional curvature.
>
> In Appendix A.2, we provide a table comparing the different operations involved with both metrics, indicating that the difference is substantial and we provide nontrivial flexibility.
>
> ---
>
> > In Line 225 - 231, what's the relationship between $f_i$ and $g_{\theta_i}$?
>
> A: Each coordinate of the output of f is the output of a $g_{\theta_i}$, i.e. $f(p) = (g_{\theta_1}(p), g_{\theta_2}(p), …, g_{\theta_k}(p))$.
>
> ---
>
> > Still in Line 225 - 231, why does $\nabla g_{\theta_i}$ is a map from M to TM?
>
> A: $\nabla g_{\theta_i}$ maps from M to TM because the gradient generates a vector field (tangent to the underlying space). If you wish to see an explicit reference, please refer to Chapter 13 of Introduction to Smooth Manifolds [33] where on page 342 (of the second edition) the gradient of $f$, a real-valued function over a smooth manifold is defined.
>
> ---
>
> > In SPD case of Line 258 - 260, since g_k does not contain any learnable parameters, how to train such a network for SPD?
>
> A: While the feature map itself has no learnable parameters, we incorporate learnable parameters from other parts of the neural network. For instance, we can learn over the extracted features $g_k$ with other neural network layers to obtain a vector field. Appendix B.2.3 describes this in detail.
>
> > How many links on average per graph used in the experiment in Section 5.1.1? It is important to verify the proposed method could perform well on medium-to-large datasets.
>
> A: We provide the total number of edges and the number of edges used in training for each of the graphs below:
>
> - Total edges
>     - Airport: 18631
>     - Disease: 2664
>     - Cora: 5278
>     - Pubmed: 44327
> - Train edges
>     - Airport: 15837
>     - Disease: 2265
>     - Cora: 4488
>     - Pubmed: 37679
>
> As can be seen, most of the edges are used as links for training. These datasets range from small/medium to relatively large (e.g. Pubmed).
>
> ---
>
> > In Table 2, the standard deviations of the proposed method is obviously larger than SPDNet, does it imply the proposed method is unstable than SPDNet?
>
> A: Although in Table 2 the standard deviations for our method are larger than those for SPDNet for the datasets of AFEW and HDM05 (arguably, they are similar for the datasets of FPHA and NTU), in Figure 5 (referenced in Appendix C.8), we find that our proposed method converges faster and to a higher value than SPDNet, despite sometimes experiencing increased standard deviation. In particular, even for HDM05, which has higher standard deviation in the table, this is exceptionally clear.
>
> ---
>
> > Will the code be released?
>
> A: We will release code for our experiments.

---

> > ### Comment · Reviewer_5HC1 · 2023-08-20
> >
> > Thanks for the authors' response. It addressed my concerns. As the authors promised to release their code, I will keep my original rating.

---

### Official Review · Reviewer_1vkd · 2023-07-05

**Soundness:** 3 good
**Presentation:** 3 good
**Contribution:** 3 good
**Rating:** 6
**Confidence:** 4

**Summary:**

The paper generalizes the ResNet layer to non-euclidean geometries by replacing the Euclidean sum with the exponential map. The theory is general and applies to any smooth metric. They propose a way to parametrize a vector field on the manifold which is more geometrically principled than the trivial vector field embedding. Empirically they show improvement in performance on hyperbolic datasets and PSD spaces compared to some baselines.

**Strengths:**

The paper theory is general and applies to any smooth manifold metric. In ResNet a neural network produces a vector field and the output of the ResNet is the input plus such vector field. The proposed generalization, assuming access to a vector field on the manifold, defines the output of the ResNet as the exponential map of the input in the direction of such a vector field. This is consistent with the Euclidean case.

A vector field in Euclidean space can simply be the output of a neural network, while on a manifold more care is required. Using an embedded vector field is straightforward, but, as correctly pointed out by the authors, it is not very principled geometrically. The authors propose a computationally tractable and geometrically principled way of defining a parametric vector field on a manifold. The idea is to make use of a collection of $\mathbb{R}$ valued functions (obtained as a projection on hyperplanes or similar) to define, through push forward and pullback, a vector field on the manifold.

Extensive specific examples of such a collection of functions are given for the hyperbolic and PSD manifolds. Experiments are also performed in these cases, and the proposed approach appears to outperform the current state of the art.

In the general manifold case (Appendix B.5), the idea of projection on pseudo-hyperplane is appealing and well-argued. And the further generalization to “hyper-disks” allows proper formal extension also to the non-geodetically-complete manifolds.

**Weaknesses:**

The definition of the vector field (Appendix B.4) is not sufficiently formal and contains mistakes. Specifically, the author assumes access to a smooth function $f:M\rightarrow \mathbb{R}^k$, a so-called “feature map”. The differential $D_x f$ is then a linear map from the tangent space in $x$, $T_x M$ to the tangent space in $f(x)$, $T_{f(x)} R^k = R^k$. Observing that the dual of $R^k$ is isomorphic to $R^k$ itself, the pullback of the differential $(D_x f)^*$ can be seen as a map from $R^k$ to $(T_x M)^*$, for every $x\in M$. This function is evaluated in $f(x)\in R^k$ such that the vector field (as defined in line 215) is a map
$$ l_f: x \rightarrow (D_x f)^*(f(x)) $$ which, as we saw, take values in $(T_x M)^*$ and NOT on $T_x M$, as line 215 is saying.

We would like to see this inconsistency explained. Is this based on the observation that both $(T_x M)^*$ and $T_x M$ are isomorphic to $\mathbb{R}^{dim(M)}$?

And also, can you reason about the choice of evaluating the pullback of the differential (line 215) in $f(x)$? Is this somehow principled?

The equivalence with Euclidean ResNet shown in Appendix D is a proper extension in the case of an embedded vector field, but it is rather difficult to follow in the case of the feature map. Specifically, Proposition 1 is trivially proved for an embedded vector field, but the same argument should also apply to the case of a feature-map-induced vector field. It would be helpful if the discussion on $g_{w,b}$ could e.g. focus on the case of axis-aligned planes (i.e. each $w$ should be an element of a standard basis and $b = 0$), such that the differential $D_x f$ reduces to an identity. We found this to be significantly more intuitive.

In the general manifold case (Appendix B.5), the idea of projection on pseudo-hyperplane is, although well explained, not at all investigated. First, it is not clear how to practically implement such projections of these pseudo-hyperplanes (not in the geodesically complete case, and even worse in the general case). Second, there are no experiments regarding general manifolds, and there is also no dissertation on the increased computational complexity with respect to hyperbolic and PSD cases. This reduces “contribution 3” in the statement in lines 72-76.

*Minor:*
* The related work section argues that the proposed construction is different from a neural ODE (which also generalizes ResNets), but honestly, we found this argument to be incomplete. It does seem like the proposed construction is a neural ODE.
* Line 126, we found it unclear which "specific structure" is being exploited in methods using Frechet means (these averages apply on practically all manifolds).

*Regarding the score:*
We are willing to increase the score if these concerns are appropriately discussed in the rebuttal.

**Questions:**

Please reply to the weaknesses above.

Furthermore:

Major:
* Line 215 and Appendix B4. The function $l_f(x)$ is inconsistent. In the main paper is a map $M \rightarrow T M$ while in the appendix is a map $M \rightarrow (T M)^*. Can you explain the inconsistency?

Minor:
* Line 189 $f_nn$ should be $f$?
* The Log-Euclidean metric amounts to a Euclidean metric in the tangent space at the identity. In that case, does $f$ reduce to being $f: T_I M \rightarrow R^k$? If so, is your construction just a standard Euclidean ResNet in a pre-specified tangent space? That would be good to state explicitly, if so.
* It's quite common to run into numerical instabilities (esp. on hyperbolic manifolds). Is that something you face?

**Limitations:**

The parting "Limitations" paragraph discusses a relevant assumption. However, we feel that a discussion of the tractability of (projections onto) "pseudo-hyperplanes" (Eq. 17 in the appendix) is lacking.

---

> ### Author Rebuttal · Authors · 2023-08-10
>
> Thank you for the review and the constructive comments! We address your comments from the “Weaknesses” and “Questions” sections below.
>
> ---
>
> > The definition of the vector field... are isomorphic to $\mathbb{R}^{\dim(M)}$?
>
> A: Since our manifold is equipped with a Riemannian metric, there is a canonical isomorphism, induced by the metric, between $(T_x M)^*$ and $T_x M$. Composing the differential operator $(D_x f)^* : R^k \rightarrow (T_x M)^*$ with this yields the map that we notatate as $(D_x f)^*_r : R^k \rightarrow T_x M$ in Appendix B.4, which we precompose with $f$ to obtain $\ell_f : M \rightarrow TM$. For more information about the isomorphism between $(T_x M)^*$ and $T_x M$ we encourage the reader to consult “The Tangent-Cotangent Isomorphism” section from chapter 13 of [33].
>
> For simplicity, by abuse of notation we overload $(D_x f)^*$ to mean $(D_x f)^*_r$ in the main paper, and point the reader to appendix B.4 for details. However, we agree that this is confusing and will add details to the main paper about this step, thereby resolving any inconsistencies.
>
> ---
>
> > And also, can you reason... somehow principled?
>
> A: We can describe the idea here, more broadly. The differential provides a natural way to map from $T_p M$ to a Euclidean space (since $T_{f(p)} R^k$ is Euclidean). We seek a natural map into the tangent space, so we take the pullback to obtain a natural map from $R^k$ into the dual space $(T_p M)^*$, and then dualize (i.e. use the tangent-cotangent isomorphism) to obtain a map from $R^k$ into $T_p M$, as desired.
>
> One motivation is that in the Euclidean case, because the maps $f(x)$ and $(D_x f)^*$ are linear, $\ell_{f}$ will reduce to a standard linear layer, which, in combination with the Euclidean $\exp$ map, will produce a standard Euclidean residual neural network.
>
> ---
>
> > The equivalence with... more intuitive.
>
> A: Yes, you are precisely correct. You can generalize the argument in Proposition 1 to the case of feature map-induced vector fields, by noting that if feature maps are projections to standard axis-aligned planes, the same reduction happens as that which was shown to hold in Proposition 1 for embedded vector fields.
>
> ---
>
> > In the general manifold... the statement in lines 72-76.
>
> A: The idea of the general manifold case feature projection is proposed mostly as a natural theoretical extension of the feature projection theory we had been developing up until that point. Please note that we tested this approach experimentally in Appendix C.4 for the hyperbolic case, and compared the performance of these pseudo-hyperplanes with the horosphere-projection based feature map approach. However, in general, applying this method may require more specific investigation of the manifold over which optimization is occurring (i.e. there may be an easier approach to obtaining geodesic distances than explicitly solving the geodesic equations).
>
> On a separate note, our claim stated in lines 72-76 refers to the ability to change the metric on the same manifold, as done in the SPD experiments in Section 5.2.
>
> ---
>
> > (minor) The related work section... is a neural ODE.
>
> A: First, it is worth pointing out that the construction is quite different from a neural ODE, in that the vectors exist on the tangent space of a manifold, not just in $R^n$. Second, speaking of neural manifold ODEs, for which the residual vectors exist in tangent spaces, our construction is effectively a generalized neural manifold ODE. Looking more closely at a manifold ODE [36, page 4], we see the neural network depends on time and generates a “flow”. A riemannian ResNet requires only the provision of a vector field, entirely untethered from any notion of solving an ODE/untethered from a time variable. This makes it a strict generalization, suitable for use in a general manifold neural network context.
>
> ---
>
> > (minor) Line 126, we found it... all manifolds.
>
> A: We meant mostly that the Helgason-Fourier construction exploits a fairly particular structure, but we also believe it is worth noting that weighted Frechet means are specifically introduced for convolution, which is not the focus of our work (we focus on residual connections). We will augment the writing to make this clear.
>
> ---
>
> > (question, major) Line 215... explain the inconsistency?
>
> A: By the tangent-cotangent isomorphism, we pass from $(T_p M)^*$ to $T_p M$. Please see our above comments regarding this for more details. The map we actually use in practice goes from $M$ to $TM$. We will make this explicit for clarity.
>
> ---
>
> > (question, minor) Line 189 f_n n should be f?
>
> A: Yes, thank you for pointing out this typo. This will be fixed.
>
> ---
>
> > (question, minor) The Log-Euclidean metric... if so.
>
> A: For the embedded vector field, the answer is yes. This is one shortcoming we observed with SPDNet, which only operates with the Log-Euclidean metric, motivating our additional use of the affine-invariant metric. For the feature map-induced vector field, where the feature maps are eigenvalue projections, the answer is no. Learning happens by way of repeated spectral remapping.
>
> ---
>
> > (question, minor) It's quite common... you face?
>
> A: This is something we have encountered, but there are ways of limiting the effect of such instabilities on the results. In particular, for the Poincare ball model we use, instabilities occur at large distances away from the origin, where $r \in (1-\epsilon, 1)$ for $\epsilon < 0.001$. As a consequence, we limit the hyperbolic distances and do projections in order to keep our optimization numerically stable. But an approach that is fully naive will likely run into some numerical difficulties, as you suggested.
>
> ---
>
> > (limitations concern) The parting "Limitations" paragraph... is lacking.
>
> A: We will add a separate discussion about tractability of pseudo-hyperplanes. We give this construction as an appealing theoretical generalization, but application to particular manifolds will require some care.

---

> > ### Comment · Reviewer_1vkd · 2023-08-14
> > **Thanks for the rebuttal**
> >
> > We thank the author for the reply. However, we would like to ask for further clarification.
> >
> > > One motivation is that in the Euclidean case, because the maps ... will produce a standard Euclidean residual neural network.
> >
> > This is very interesting and can be a sufficient motivation for the choice of evaluation on $f(x)$. Can the author provide step-by-step reasoning and proof of this statement? Why will $l_f$ reduce to a standard linear layer?

---

> > > ### Author Response · Authors · 2023-08-15
> > > **Further clarification**
> > >
> > > We can certainly do so. Note for the Euclidean case, that our feature map $f : \mathbb{R}^n \rightarrow \mathbb{R}^k$ will, for example ($b=0$, $W$ has normalized row vectors), take the form $f(x) = Wx, W \in \mathbb{R}^{k \times n}$. Then note that we have $Df = W$ and $(Df)^* = W^T$. We see for the standard feature map-based construction, our vector field $\ell_f (x) = (D_x f)^* (f(x))$ takes the form $\ell_f (x) = W^T W x$.
> > >
> > > For the learnable case (which is standard for us, given that we learn Riemannian residual neural networks), note from Lines 217-219 that we have $\ell_{f,\theta} (x) = (D_x f)^* (n_\theta (f(x)))$ for $n_\theta$ a neural network. Hence we have $\ell_f (x) = W^T n_\theta (W x)$. For the case that you mentioned before, i.e., when the feature maps are trivial projections (onto axis-aligned hyperplanes), we have $W= I$ and $\ell_f (x) = n_\theta(x)$. Thus our construction can be viewed as a generalization of a standard neural network.

---

> > > > ### Comment · Reviewer_1vkd · 2023-08-16
> > > > **Increasing score**
> > > >
> > > > Thank you for the follow-up.
> > > >
> > > > This should settle our last concerns, and we will increase our score to recommend acceptance.
> > > >
> > > > Should the paper get accepted, we hope that the above discussions will be incorporated into the final paper.

---

> > > > > ### Author Response · Authors · 2023-08-20
> > > > > **Thank you**
> > > > >
> > > > > We would once again like to thank you for your detailed comments and your time! We will incorporate these discussions into the paper and ensure the paper is clear with respect to the above mentioned details.

---

### Official Review · Reviewer_1LjF · 2023-07-10

**Soundness:** 3 good
**Presentation:** 2 fair
**Contribution:** 2 fair
**Rating:** 6
**Confidence:** 2

**Summary:**

The paper proposes an extension of standard ResNets called Riemannian Residual Neural Networks. The extension is done based on Riemannian manifolds as discussed in Equation (2). Some numerical results on node classification problems are presented in section 5 to show the improvements of the proposed generalization of ResNets.

-- Post-rebuttal Review Update --

I thank the authors for the detailed responses to my comments. I find the responses satisfactory and raise my score to 6.

**Strengths:**

1- The idea of Riemannian ResNets sounds interesting and as the numerical results suggest could help improve the performance of ResNet models in applications where the chosen Riemannian geometry suits the dataset.

**Weaknesses:**

1- The paper's presentation remains abstract in the main body, and I do not find the current presentation accessible enough to deep learning practitioners. For example, Section 3 spends about 2.5 pages explaining Riemannian geometry but does not discuss a concrete example where the exponential map and vector fields can be discussed. The examples in section 4.2.1 appear late in the draft and also do not derive the expression for the exponential map that appears in Riemannian ResNets.

2- Since the paper has not discussed the algorithmic steps of training and evaluating a Riemannian ResNet, it is not that easy to see how the network can be trained for non-Euclidean Riemannian geometries. I suggest adding one or two algorithms to the draft to discuss the steps of training a Riemannian ResNet for the cases discussed in Section 4.2.1.

**Questions:**

How did the computational costs of training a Riemannian ResNet compare to that of a standard ResNet? Could the Riemannian ResNet demand more computational power for training than the normal ResNet?

**Limitations:**

Please see my previous responses.

---

> ### Author Rebuttal · Authors · 2023-08-08
>
> Thank you for the review and the constructive comments. We appreciate that you think our idea is interesting and has the potential to improve the performance of ResNets over datasets with chosen Riemannian geometry. We address your comments from the “Weaknesses” section below as well as your questions from the “Questions” section.
>
> ---
>
> > The paper's presentation remains abstract in the main body, and I do not find the current presentation accessible enough to deep learning practitioners. For example, Section 3 spends about 2.5 pages explaining Riemannian geometry but does not discuss a concrete example where the exponential map and vector fields can be discussed. The examples in section 4.2.1 appear late in the draft and also do not derive the expression for the exponential map that appears in Riemannian ResNets.
> ‎
> Since the paper has not discussed the algorithmic steps of training and evaluating a Riemannian ResNet, it is not that easy to see how the network can be trained for non-Euclidean Riemannian geometries. I suggest adding one or two algorithms to the draft to discuss the steps of training a Riemannian ResNet for the cases discussed in Section 4.2.1.
>
> A: We apologize for any confusion. Due to page limits, we decided to offload these details to the appendix. We would like to refer the reader to the Appendix where we extensively document how to implement a Riemannian ResNet:
>
> 1. In Appendix A, we provide a table of the expressions we use to compute the exponential map on various manifolds. All operations can be performed with standard PyTorch operations.
> 2. In Appendix B, we elaborate on the Riemannian ResNet design presented in Section 4, discussing the tradeoffs of each design.
> 3. Appendix C outlines more experimental details. In Appendix C.7, we give a concrete example of how we constructed a Riemannian ResNet for covariance matrix classification.
>
> We will include more of these details in the main body in a revised version.
>
> ---
>
> > How did the computational costs of training a Riemannian ResNet compare to that of a standard ResNet? Could the Riemannian ResNet demand more computational power for training than the normal ResNet?
>
> A: One of the key benefits of a Riemannian ResNet is that it can capture geometric invariants of the training data. This can reduce the parameter count of the neural networks used, improving both performance and  efficiency. For example, because the Riemannian ResNet for SPD matrices acts on eigenvalues, it is invariant to change of eigenbasis. While the size of a $n \times n$ matrix grows at an $O(n^2)$ rate, the number of eigenvalues grows at only an $O(n)$ rate, leading to computational efficiency. We demonstrate these benefits in Appendix C.10, where Euclidean ResNet performs worse than the Riemannian ResNet across all SPD datasets. However, computation can be a challenge when closed-form solutions to the exponential maps are unknown.
>
> We believe that our work can highlight these computational benefits of geometric machine learning, and will clarify this in a revised version.

---

> ### Author Response · Authors · 2023-08-20
> **Follow-up**
>
> Dear reviewer,
>
> We would like to ask if we have addressed your concerns, and if so, if it would be possible to raise your rating for the paper?
>
> If any additional questions have arisen, please let us know.

---

### Decision · Program_Chairs · 2023-09-21

**Decision:**

Accept (poster)

**Comment:**

The paper deals with ResNets and how to deal with manifold structures there. While some reviewers found the paper to be a natural extension to existing works, others found the paper interesting for a subset of the NeurIPS community. Overall, after discussions, I am of the opinion that the paper merits acceptance with the hope that many of the suggestions should be incorporated.